# To Think or Not To Think: A Study of Thinking in Rule-Based Visual Reinforcement Fine-Tuning

**Ming Li[1]**  **Jike Zhong[2]**  **Shitian Zhao[1]**  **Yuxiang Lai[3]**  **Haoquan Zhang[1, 4]**  **Wang Bill Zhu[2]**
**Kaipeng Zhang[1]** [*]

[1]Shanghai AI Laboratory
[2]University of Southern California
[3] Emory University
[4] Chinese University of Hong Kong
lm1640362161@gmail.com, zhangkaipeng@pjlab.org.cn

Project code: https://github.com/minglllli/CLS-RL

## Abstract

This paper investigates the role of explicit thinking process in rule-based reinforcement fine-tuning (RFT) for multi-modal large language models (MLLMs). We first extend *Thinking-RFT* to image classification task, using verifiable rewards for fine-tuning (FT). Experiments show Thinking-RFT significantly outperforms supervised FT and yields a cross-dataset generalization effect. We then rethink and question whether explicit thinking in RFT is always necessary and beneficial. Challenging the convention that explicit thinking is crucial for the success of RFT, we introduce *No-Thinking-RFT*, exploring RFT without thinking by introducing a simple equality accuracy reward. We evaluate No-Thinking-RFT on six diverse tasks across different model sizes and types. Experiment results reveal four key findings: **(1).** Visual perception tasks do not require thinking during RFT, as No-Thinking-RFT consistently outperforms or matches Thinking-RFT across model sizes and types. **(2).** Models with limited capabilities struggle to generate high-quality CoT for RFT, making Thinking-RFT less effective than No-Thinking-RFT. **(3).** There are inconsistencies between the answers in the thinking tags and answer tags for some responses of Thinking-RFT, which show lower average accuracy than the overall accuracy. **(4).** The performance gain of No-Thinking-RFT mainly stems from improved learning during no thinking FT and the avoidance of inference overthinking, as evidenced by the partial gains from appending empty thinking tags at inference time of Thinking-RFT. We hypothesize that explicit thinking before verifiable answers may hinder reward convergence and reduce performance in certain scenarios. To test this, we propose *Think-After-Answer*, which places thinking after the answer to mitigate this effect for experimental verification. Lastly, we conduct a pilot study to explore whether MLLMs can learn when to think during RFT, introducing an *Adaptive-Thinking* method. Experiments show that model converges to either thinking or not depending on model capability, achieving comparable or better performance than both Thinking and No-Thinking-RFT. Our findings suggest MLLMs can adaptively decide to think or not based on their capabilities and task complexity, offering insights into the thinking process in RFT.

## 1 Introduction

Recently, rule-based reinforcement fine-tuning (RFT) has made significant progress and achieved better performance than traditional supervised fine-tuning (SFT) [19, 68, 26]. RFT leverages verifiable

---

[*]Corresponding Author: zhangkaipeng@pjlab.org.cn

39th Conference on Neural Information Processing Systems (NeurIPS 2025).

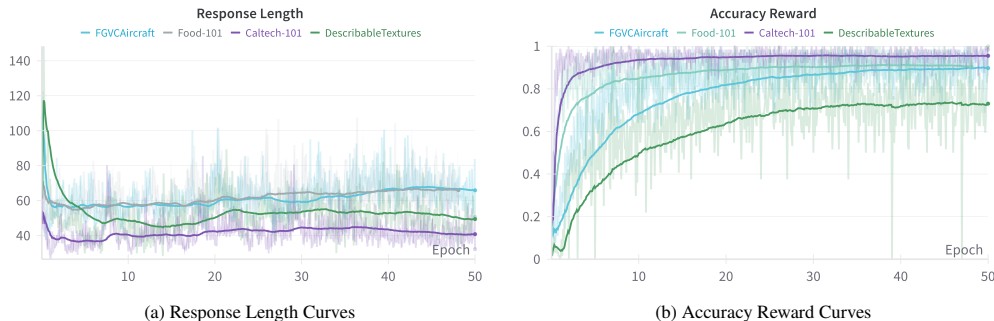

(a) Response Length Curves     (b) Accuracy Reward Curves

Figure 1: **Thinking is less essential**: Classification RFT exhibits sharp drops in response length at specific steps, accompanied by significant rises in accuracy reward.

rewards for training, encouraging models to engage in a thinking process before answering for solution exploration [19]. The explicit thinking is widely believed as a key factor in RFT's success, and many works on multi-modal RFT [82, 24] aim to reproduce the length-increasing and 'aha moment' effects seen in Deepseek-R1 [19]. However, a critical question emerges as RFT being widely used: *Is explicit thinking always necessary and beneficial for small-sized-model RFT?* Recent studies [27, 64, 65] suggest that reasoning offers limited gains on commonsense tasks and overthinking may even harm inference performance. While these findings offer insight into model reasoning, they focus *solely on inference*, leaving the *impact of explicit thinking during RFT unclear and largely unexplored*. Furthermore, RFT typically requires significantly more fine-tuning time and GPU memory than SFT due to the generation of multiple lengthy responses [19]. Consequently, the role of explicit thinking during RFT warrants further exploration, considering both accuracy performance and computational efficiency.

In this paper, we investigate the thinking process in rule-based RFT for MLLMs across different tasks and model sizes. We begin with a case study exploring closed-form MLLM image classification. Motivated by the success of rule-based RFT [19, 68, 77] in LLM fine-tuning, we extend thinking-based RFT (Thinking-RFT) to few-shot classification fine-tuning. Thinking-RFT fine-tunes MLLMs using class labels as verifiable answers for reward calculation. Extensive experiments show that Thinking-RFT performs much better than SFT on both in-domain learning and new-class generalization. Additionally, we observe a *free-lunch phenomenon*: fine-tuning MLLMs on one dataset with Thinking-RFT improves performance on other datasets, despite shift in data distribution and entirely different class names. This phenomenon validates that rule-based RFT can effectively teach models the fundamentals of image classification rather than simply memorizing [8].

We then revisit the role of thinking process in rule-based RFT, a key factor in the success of Deepseek-R1 [19]. Unlike the gradual increase in response length of math problems observed in [19], classification RFT exhibits sharp drops in response length at specific steps, accompanied by significant rises in accuracy reward (Figure 1). Our finding that thinking is less essential for classification, prompting models to adopt minimal reasoning, aligns with recent studies [27, 64] showing that overthinking during inference can hinder performance on certain tasks. To this end, we propose No-Thinking-RFT, a rule-based RFT approach without explicit thinking process. No-Thinking RFT utilizes a direct-answer instruction prompt and removes the format reward. It implements a strict equality accuracy reward, granting a score of 1 solely when the output precisely matches the labels, thereby effectively discouraging reasoning during fine-tuning. Notably, in few-shot classification task, No-Thinking-RFT outperforms Thinking-RFT while achieving substantially shorter fine-tuning and inference time (Sec.4.4) and faster convergence (Sec.4.5) than Thinking-RFT.

Next, we evaluate No-Thinking-RFT on general visual reasoning tasks, including math, spatial reasoning, puzzles, referring grounding, and detection, across different model sizes. Results reveal four important findings: **(1):** Thinking is unnecessary for visual perception tasks during RFT. Across all test model sizes, No-Thinking-RFT matches or outperforms Thinking-RFT for these tasks, showing that excluding thinking in RFT can preserve or improve performance while enhancing training and inference efficiency. **(2).** Models with limited capability (e.g., 2B) tend to converge to produce trivial reasoning under Thinking-RFT (refer to Figure 15 ∼ Figure 20 for examples), leading to worse performance and longer fine-tuning time compared to No-Thinking-RFT. **(3):** For mid-sized models

(e.g., 7B), Thinking-RFT often produces inconsistencies between the content in thinking tags and answer tags (see Figure 9∼Figure 14). We observe frequent mismatches comparing answers from the thinking tags and answer tags, with the average accuracy amongst inconsistent responses lower than the overall accuracy (Figure 2), which suggests that maintaining response consistency could improve performance. **(4):** The performance gain of No-Thinking-RFT over Thinking-RFT mainly stems from improved fine-tuning and the avoidance of inference overthinking. When we test Thinking-RFT in a no-thinking mode by appending empty thinking tags during inference, performance improves on perception tasks but still lags behind No-Thinking-RFT. This suggests that both enhanced learning during no thinking fine-tuning and reduced inference overthinking contribute to the gain.

We further study why No-Thinking-RFT outperforms Thinking-RFT in certain scenarios (e.g., 2B model on perception tasks), hypothesizing that the explicit thinking before verifiable answers may hinder the learning process during RFT, slowing accuracy reward convergence and reducing accuracy. To test this, we propose *Think-After-Answer*, a variant where reasoning occurs after generating verifiable answers. Experiment results support our hypothesis: in tasks where No-Thinking-RFT outperforms Thinking-RFT, Think-After-Answer converges faster and achieves higher accuracy than Thinking-RFT. However, it still underperforms No-Thinking-RFT, suggesting that reasoning, even when placed after answers, can remain detrimental during RFT in certain scenarios.

Finally, we conduct a pilot study to investigate whether MLLMs can learn to adaptively decide when to think during RFT. We introduce *Adaptive-Thinking*: models are prompted to first assess if a problem required reasoning or thinking. If reasoning was deemed necessary, the model would generate a thinking process before answering; otherwise, it would answer directly. Our experiments revealed that models consistently converged to a single response strategy (either always thinking or never thinking). Notably, Adaptive-Thinking achieves comparative or better performance than the better one of Thinking-RFT and No-Thinking-RFT, and the final response strategy always corresponded to one suited for that specific model size and task complexity. For example, on math tasks, the 2B model converged to the non-thinking answering response, whereas the 7B model consistently adopted the thinking process before answering. These findings suggest that MLLMs may possess the capability to learn whether to employ reasoning based on their inherent abilities and task complexity during RFT.

Our contributions can be summarized as follows:

- We extend Thinking-RFT to few-shot MLLM classification and reveal a cross-dataset transfer across datasets.
- We show several important findings about No-Thinking-RFT and Thinking-RFT via experiments on six tasks with 2B–7B models.
- Through Think-After-Answer and Adaptive-Thinking we confirm that deferring or omitting CoT speeds convergence without harming accuracy and explore adaptive thinking strategy.

## 2 RFT Methods: Think *vs.* No Think

### 2.1 Thinking-RFT

**Optimization Algorithm.** We follow Deepseek-R1 [19, 59] to employ Group Relative Policy Optimization (GRPO) as the RL algorithm for optimization in our study, since it is the most widely used one. We refer readers to Appendix A for a brief introduction of the technical details of GRPO.

**Instruction prompt.** Following Deepseek-R1 [59], we utilize a prompt that encourages models to first engage in a thinking process before generating the final answers. The prompt is designed as: *{Question} Please output the thinking process in <think> </think> and final answer in <answer> </answer> tags.* Here *{Question}* will be replaced by each specific question.

**Reward Function.** To clearly examine the RFT thinking process and promote the generalizability of our findings, we employ the simplest binary reward function for our study. The reward function is composed with two parts: format and accuracy reward. The format reward $R_{\text{format}}$ is to check if the responses follow the format correctly. $R_{\text{format}} = 1$ if the response format is correct, and 0 if it is incorrect. The accuracy reward $R_{\text{accuracy}}$ checks whether the answer in the answer tag matches the ground truth. For example, $R_{\text{accuracy}}$ verifies the extracted answer against the correct choice in multi-choice problems, class labels in classification tasks, and numeric results in math problems. $R_{\text{accuracy}} = 1$ if the extracted answer matches correctly, and 0 otherwise.

**Training Strategy.** To study explicit thinking during RFT clearly, we adopt R1-zero [19] training for all methods in our study, i.e., we apply RL to all base models without any SFT, following [80].

## 2.2 No-Thinking-RFT

We explore rule-based RFT without a thinking process and propose a No-Thinking-RFT method. The instruction prompt and reward functions are designed as below:

**Instruction prompt.** Instead of the prompt in Thinking-RFT which encourages models to think before answering, the prompt in the No-Thinking-RFT method prohibits the model from thinking. The prompt is designed as: {Question} Please directly output the answer.

**Reward Function.** No-Thinking-RFT eliminates the format reward and relies solely on the accuracy reward rather than combining two rewards. The accuracy reward $R_{\text{accuracy}}$ checks whether the model's output matches the ground truth exactly. $R_{\text{accuracy}} = 1$ if the model response matches the ground truth and 0 otherwise. The equality-based matching reward forces the model to bypass any thinking process and output only the answers, which are significantly shorter than the typical reasoning responses in Thinking-RFT. As a result, training No-Thinking-RFT requires substantially less GPU memory, and its training and inference times are markedly shorter than those of Thinking-RFT (Sec. 4.4).

## 3  General Visual Reasoning

In this section, we introduce results on general visual reasoning, including spatial understanding, math, puzzle, referring grounding, and detection across different model sizes. We report the results of spatial understanding, math, and puzzle tasks with 2B and 7B Qwen2-VL-Instruct [72] models in main text, and leave the results of referring grounding, detection tasks and other different model types (e.g., InternVL2.5 [5] and Qwen2-5-VL-Instruct [1]) in Appendix G. We also discuss offline DPO, No-Thinking-RFT, and Thinking-RFT in Appendix H.2. We first introduce Think-After-Answer and Adaptive-Thinking methods and then report experimental results.

### 3.1  Think-After-Answer & Adaptive-Thinking

**Think-After-Answer.** We explore the reason that No-Thinking-RFT outperforms Thinking-RFT under specific tasks and model sizes. As shown in Sec. 4.5, Thinking-RFT converges slower than No-Thinking-RFT. Therefore, we hypothesize that for the 2B model, the explicit "think" process is detrimental, whereas for the 7B model, it is not universally required, as verifiable answers are already conditioned on thinking. To verify this hypothesis, we propose Think-After-Answer by letting MLLMs first answer the questions and then output a brief reasoning process, therefore alleviating the negative impact of explicit thinking on verifiable answers during RFT, as now the thinking is conditioned on answers. The prompt is provided in Appendix E.2 and the format and accuracy reward do not change. If the hypothesis holds, the convergence speed of Think-After-Answer should be faster and the final performance should be higher than Thinking-RFT in cases where No-Thinking-RFT outperforms Thinking-RFT.

**Adaptive-Thinking.** We propose an Adaptive-Thinking method that lets MLLMs decide whether to think by themselves during RFT. Specifically, we prompt MLLMs to first determine whether a problem requires thinking, and then either output a reasoning process if needed or directly output an answer if not (refer to Appendix E.3 for the prompt). A response in either thinking format or direct answering format will receive a format reward of 1. The accuracy reward is not changed as before.

### 3.2  Experimental Setup

We focus on three main visual or multi-modal tasks: visual perception (classifiction and spatial grounding), multi-modal math reasoning, and visual puzzle reasoning in the main text. For a fair comparison, we set hyperparameters the same for all methods. We provide a brief overview of the experimental settings below, with detailed implementation described in Appendix D.3.

**Visual Perception.** We follow [82] to fine-tune models on SAT dataset [57] 2 epochs and then test on CVBench dataset [69]. We also include classification results for comparison. The number of rollout

Table 1: Results of Thinking-RFT, Think-After-Answer, No-Thinking-RFT, and Adaptive-Thinking on CVBench.

| Model | Method | Overall | Count | Relation | Depth | Distance | FT Time |
|-------|--------|---------|-------|----------|-------|----------|---------|
| 2B | Thinking-RFT | 70.36 | 66.12 | 83.38 | 68.50 | 63.67 | 599 m |
|  | Think-After-Answer | 73.65 | 68.65 | 82.15 | 74.83 | 69.83 | 408 m |
|  | No-Thinking-RFT | 76.76 | 69.67 | 84.46 | **80.67** | 73.83 | **139 m** |
|  | Adaptive-Thinking | **77.03** | **69.92** | **86.31** | 76.50 | **76.83** | 208 m |
| 7B | Thinking-RFT | 80.36 | 66.24 | 90.92 | 87.33 | 80.00 | 651 m |
|  | Think-After-Answer | 81.61 | 66.75 | 90.46 | 87.33 | **85.83** | 428 m |
|  | No-Thinking-RFT | 80.67 | 66.50 | **92.15** | 83.83 | 83.67 | **155 m** |
|  | Adaptive-Thinking | **81.65** | **67.38** | 90.31 | **88.00** | 84.67 | 639 m |

Table 2: Performance comparison of PuzzleVQA and AlgoPuzzleVQA across 2B and 7B models.

| Method | 2B | | 7B | |
|--------|----------|---------------|----------|---------------|
|  | PuzzleVQA | AlgoPuzzleVQA | PuzzleVQA | AlgoPuzzleVQA |
| Thinking-RFT | 52.50 | 27.72 | 66.60 | 24.78 |
| Think-After-Answer | 64.70 | 26.94 | 80.45 | 28.11 |
| No-Thinking-RFT | 70.85 | **29.17** | 80.65 | **29.39** |
| Adaptive-Thinking | **75.45** | 27.94 | **85.05** | 29.00 |

is set to 4 for both 2B and 7B models. The number of rollout is set to 4 for both 2B and 7B models. $\beta$ is set to 0.04, learning rate is set to $1 \times 10^{-6}$. The temperature is set to 1 for all experiments.

**Multi-Modal Math Reasoning.** We utilize the Math-40K [61] as the find-tuning data and fine-tune models 1 epoch. Then we test the fine-tuned models on both MathVista [43] and MathVision [71]. The maximum length of input prompt and response are set to 4096 and 512 respectively. The number of rollout is set to 8 for 2B model, and 4 for 7B model. $\beta$ is set to 0.04, learning rate is set to $1 \times 10^{-6}$.

**Visual Puzzle Reasoning.** We follow the code of [7] to generate a training dataset with 6.5k data and fine-tune models 2 epochs. We then test fine-tuned models on PuzzleVQA [7] as in-domain testing, and on AlgoPuzzleVQA [17] as out-of-domain testing. $\beta$ is set to 0.04, learning rate is set to $1 \times 10^{-6}$. The batch size is set to 1 per GPU and we use 2-step gradient accumulation during training.

### 3.3 Results & Findings

We compare the results of four different thinking strategies here and show examples of responses in Figure 21~Figure 25. We report the results of visual perception on Table 1, math reasoning on Table 3, and puzzle reasoning on Table 2. We analyze these results and discuss several findings below.

**Finding 1: For small models with weak ability (e.g., 2B size), Thinking-RFT may lead to trivial reasoning and inferior performance compared with No-Thinking.** We observe that for 2B models, by using much less fine-tuning time, No-Thinking-RFT outperforms Thinking-RFT in most benchmarks by a large margin, including even Mathvista and except only MathVision. We find that for Mathvista, No-Thinking-RFT outperforms Thinking-RFT on all tasks except math world problems (MWP). It is reasonable that Thinking-RFT outperforms No-Thinking-RFT in MWP and MathVision since these problems need much computation for obtaining final answers and it is hard to directly output correct answers. When looking deeper into model responses under Thinking-RFT, we find that most reasoning is trivial, i.e., it does not contribute meaningfully to the final answer, especially for complex reasoning tasks (refer to Figures 15~Figure 20 for qualitative examples). These results suggest that when the capability of models is poor, the thinking process during RFT may converge to trivial solutions and finally lead to inferior performance compared with No-Thinking.

**Finding 2: Visual perception and puzzle tasks do not need thinking.** We find that No-Thinking-RFT outperforms Thinking-RFT on spatial reasoning, classification (7B results in Table 12), and puzzle tasks across both 2B and 7B models, while Thinking-RFT performs better on math reasoning with the 7B model. We notice that although puzzle tasks typically require complex reasoning, No-Thinking-RFT outperforms Thinking-RFT on both 2B and 7B models. This may be because these tasks rely on visual rather than linguistic reasoning, and language-based reasoning might cause

Table 3: Accuracy (%) of 2B and 7B models on MathVista sub-tasks and MathVision overall. TQA: Textbook QA, VQA: Visual QA, Geo: Geometry, MWP: Math Word Problem, FQA: Figure QA.

| Model | Method | MathVista | | | | | | MathVision |
| | | Overall | TQA | VQA | Geo | MWP | FQA | |
|---|---|---|---|---|---|---|---|---|
| 2B | Thinking-RFT | 44.90 | 46.84 | 36.87 | 37.02 | 52.15 | 50.19 | **16.45** |
| | Think-After-Answer | 48.50 | 51.27 | 44.69 | 36.54 | 50.54 | **57.25** | 14.80 |
| | No-Thinking-RFT | 48.80 | **52.53** | 46.37 | **38.94** | 49.46 | 55.39 | 13.49 |
| | Adaptive-Thinking | **50.20** | 50.00 | **51.96** | 37.98 | **52.69** | 56.88 | 13.16 |
| 7B | Thinking-RFT | 64.60 | **65.19** | **59.22** | **60.58** | 68.28 | **68.40** | **21.71** |
| | Think-After-Answer | 62.00 | 63.92 | 58.10 | 58.65 | 63.98 | 64.68 | 21.71 |
| | No-Thinking-RFT | 59.10 | 65.19 | 58.66 | 55.77 | 56.45 | 60.22 | 18.09 |
| | Adaptive-Thinking | **64.70** | 64.56 | 56.42 | 59.13 | **74.73** | 67.66 | 21.05 |

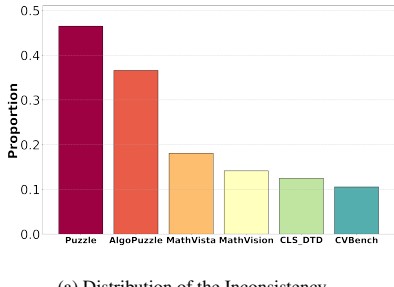

(a) Distribution of the Inconsistency

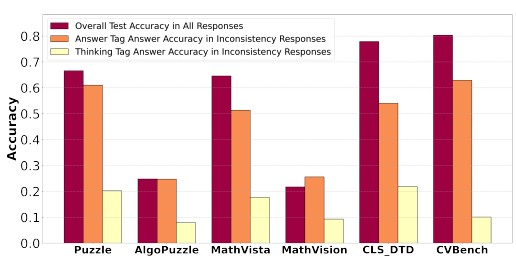

(b) Test Accuracy

Figure 2: Comparison of inconsistency proportion and test accuracy among different datasets.

hallucinations, or the 7B model's puzzle-solving capability may still be limited. In summary, visual perception and puzzle reasoning tasks do not need reasoning among test model sizes, and RFT without thinking is beneficial to both accuracy performance and computational efficiency.

**Finding 3: There are inconsistencies between thinking and answer content of some responses.** We use GPT-4o [25] to extract answers from the thinking tag and compare them with those in the answer tag (see Appendix E.1 for details). We find inconsistencies in some responses between the thinking and answer tags. Figure 2 visualizes the proportion of inconsistent responses per dataset and their respective accuracies in both tags. We also show examples of inconsistent responses in Figure 9~Figure 14. We found that **1).** Inconsistencies are more common in reasoning tasks such as puzzles than in perception tasks like classification. **2).** For inconsistent responses, accuracy of answers in answer tags is much higher than in thinking tags, and both are below the overall accuracy. These results suggest that encouraging more consistent responses could improve performance, particularly on reasoning tasks. We leave strategies for enhancing consistency to future work.

**Finding 4: Performance gain of No-Thinking-RFT over Thinking-RFT on certain tasks stems from improved learning during RFT and avoidance of inference overthinking.** We test the No-Thinking inference mode of Thinking-RFT by appending an empty thinking tag during inference and show the results on Table 17. We find that while appending empty thinking tags during inference could improve performance on CVBench and puzzle tasks, its performance is still far behind No-Thinking-RFT. This suggests that the gain of No-Thinking-RFT primarily stems from two factors: improved learning during fine-tuning and the avoidance of overthinking during inference.

**Finding 5: Explicit thinking before verifiable answers may leads to slower reward convergence and inferior performance.** We observe that Think-After-Answer outperforms Thinking-RFT in perception and puzzle tasks. We also visualize the accuracy-reward curves on the SAT and Puzzle datasets in Figure 3 (2B) and Figure 8 (7B), where Think-After-Answer achieves faster reward convergence and higher final accuracy than Thinking-RFT. These results indicate that placing explicit CoT after verifiable answers during RFT can alleviate its negative impact, improving both convergence speed and performance, thus validating our hypothesis that explicit thinking before answers may lead to slower reward convergence and reduced performance on perception and puzzle tasks.

**Finding 6: MLLMs can adaptively learn whether to think on task level during RFT on task level.** We find that Adaptive-Thinking performs comparably to or better than the stronger of Thinking-RFT

Table 4: Parameter Difference ($L_1$ Norm)

| Dataset | Method | Visual | Language | MLP_V | norm_V | attn_V | MLP_L | norm_L | attn_L |
|---------|--------|--------|----------|-------|--------|--------|-------|--------|--------|
| DTD | Thinking | 0.267 | 0.434 | 0.206 | 0.001 | 0.158 | 0.372 | 0.000 | 0.153 |
|     | No-Thinking | 0.237 | 0.402 | 0.184 | 0.001 | 0.143 | 0.379 | 0.000 | 0.154 |
| SAT | Thinking | 0.503 | 0.712 | 0.398 | 0.003 | 0.311 | 0.614 | 0.000 | 0.256 |
|     | No-Thinking | 0.518 | 0.718 | 0.398 | 0.003 | 0.327 | 0.704 | 0.000 | 0.281 |
| Math | Thinking | 0.415 | 0.735 | 0.364 | 0.002 | 0.210 | 0.691 | 0.000 | 0.270 |
|      | No-Thinking | 0.731 | 0.931 | 0.581 | 0.004 | 0.455 | 0.880 | 0.000 | 0.388 |

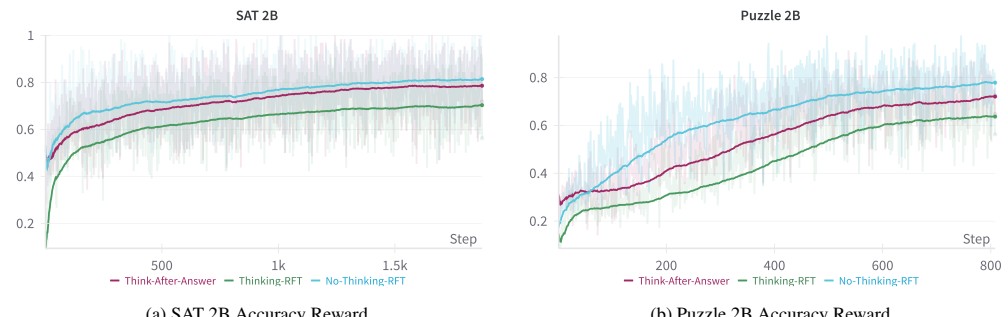

(a) SAT 2B Accuracy Reward       (b) Puzzle 2B Accuracy Reward

Figure 3: Comparison of accuracy reward convergence speed on SAT dataset over steps of Thinking-RFT, Think-After-Answer, adn No-Thinking-RFT across 2B and 7B models.

and No-Thinking-RFT. Figure 6 shows the response length trend curves during RFT, and Table 10 summarizes the statistics of test response types. We observe that models initially produce a mix of thinking and No-Thinking responses, but gradually converge to a single response type—either think or not think and matching the better-performing strategy. For instance, the 2B model converges to No-Thinking response, while the 7B model converges to thinking response for math tasks. These results suggest that MLLMs can learn whether to think during RFT based on model capacity and task complexity. Despite promising results, the same task can present problems requiring varied responses. Ideally, Adaptive-Thinking would operate at the problem level, a refinement for future work.

## 3.4 Parameter Change during RL

We compare the changes in parameters by computing the $L_1$ norm of parameter difference. Our analysis focuses on three aspects: **1. Modality**: changes between visual and language components, **2. Module**: changes in different modules, **3. Layer**: changes across different layers.

We analyzeQwen2VL-2B on SAT, DTD, and Math datasets using Thinking and No-Thinking-RFT. Modality- and Module-level results are given in the table below. We directly discuss layer-level findings as the table is too large.

**1. Modality-Level**: Language weights drift more than visual weights for all dataset—and the No-Thinking strategy raises visual drift slightly. This implies that the reward gradient mainly applies to the language modality.

**2. Module-Level**: MLP is the dominant changed component for both visual and language parts, and attention blocks change less but still contribute near 15-20%.

**3. Layer-Level**: With Thinking, weight drift grows toward deeper layers as reward back-propagates through the whole reasoning chain to high-level semantics. With No-Thinking, drift peaks in early-mid layers and then declines, indicating that low-level features are reshaped so a shallow forward path already produces the reward token.

## 4 Case Study on Image Classification Tasks

In this section, we introduce the experiment results on classification. Recent works [81] show that the classification abilities of MLLMs [42, 36, 12, 13, 49, 6] are poor due to pre-training data limitations

Table 5: No-Thinking-RFT outperforms Thinking-RFT in 10 out of 11 datasets under few-shot learning results. S.C.: StanfordCars dataset. F.A.: FGVCAircraft dataset.

| | Caltech101 | DTD | EuroSAT | Food101 | ImageNet | Flowers102 | OxfordPets | S.C. | SUN397 | UCF101 | F.A. | Average |
|---|---|---|---|---|---|---|---|---|---|---|---|---|
| Qwen2VL | 88.56 | 54.79 | 45.68 | 77.54 | 70.8 | 64.43 | 73.89 | 35.77 | 63.83 | 66.22 | 42.75 | 62.21 |
| SFT | 93.91 | 71.336 | **75.16** | 75.75 | 41.60 | **96.87** | 85.80 | 71.13 | 41.66 | 63.81 | 60.15 | 70.65 |
| Thinking-RFT | 98.09 | 69.92 | 49.46 | 88.94 | 92.24 | 86.56 | **87.24** | 80.24 | 84.57 | 82.1 | 74.41 | 81.25 |
| No-Thinking-RFT | **98.46** | **73.52** | 58.02 | **90.78** | **92.31** | 91.6 | 86.13 | **92.5** | **86.72** | **83.82** | **74.41** | **84.39** |

and SFT with large-scale data could improve performance [81]. However, acquiring large-scale datasets incurs substantial cost and computational overhead [84]. Few-shot learning thus emerges as a compelling alternative, previously validated in contrastive vision-language models (VLMs) [54, 83, 84, 38, 30], but its application to MLLMs remains underexplored. We report the results of few-shot learning, free-lunch phenomenon, convergence and efficiency comparison in this section, the results of base-to-new setting in Appendix F.1, open-set classification in Appendix F.2, the results of 7B Qwen2-VL-Instruct [72] in Appendix F.3, and examples of model response in Appendix I.

## 4.1 Experimental Setup

**Setting.** In this paper, we mainly focus on closed-form classification for MLLMs, where a subset of class names is provided for selection. The question format is {Question} {Instruction prompt}, where {Question} and {Instruction prompt} will be replaced by the specific questions and instruction prompt of each dataset and method respectively.

**Datasets.** Following [83, 84], we conduct experiments on 11 public classification benchmarks. The names, questions, statistics of each dataset are provided in Appendix C. Due to computational resource and model input limit, we select a subset of class labels as the choice list. For few-shot classification, we sample 40% of labels including the gt labels to form the choice list in question.

**Implementation Details.** We utilize Qwen2-VL-2B-Instruct [72] as the base model, and fine-tune all parameters, following [82, 3]. All training is conducted in 8 A100 GPUs. The batch size is set to 1 per GPU and we use 2-step gradient accumulation during training [3, 60]. All images are resized to 328×328 resolution and no data augmentation is used. For inference test, we first extract answers from the answer tag and then verify if class names are in answers. If the answer tag does not exist in model responses we directly verify if class names are in model responses, following [81]. The temperature is set to 1.0 for all experiments. The random seed is set to 100 for all experiments.

More details about dataset construction and implementation details are provided in Appendix D.

## 4.2 Few-Shot Learning Results

In this subsection, we present the results of few-shot learning. We train models on 4-shot setting and report the accuracy results in Table 5 for 2B models and Table 7 for 7B models. For 2B models, we observe that Thinking-RFT markedly surpasses SFT in most datasets, resulting in a notably higher average accuracy than SFT, which implies that rule-base RFT can let model learn better downstream knowledge than SFT. We further discover that No-Thinking-RFT outperforms Thinking-RFT in 10 out of 11 datasets, ultimately achieving a 3.14% higher average accuracy compared to Thinking-RFT. For 7B models, the performance gap between Thinking-RFT and No- Thinking-RFT narrows but remains. These results indicate that RFT without the thinking process can effectively enhance model's performance on classification than Thinking-RFT.

## 4.3 Free-Lunch Phenomenon

In this section, we discuss the free-lunch phenomenon. Previous work in few-shot contrastive VLM fine-tuning has demonstrated that, when fine-tuned on a specific dataset, the performance of VLMs on other datasets is drastically degraded, a phenomenon known as catastrophic forgetting [83, 30]. However, we will show that Thinking-RFT and No-Thinking-RFT can enhance the performance of MLLMs on other datasets when fine-tuned on one specific dataset. We visualize the improvement of

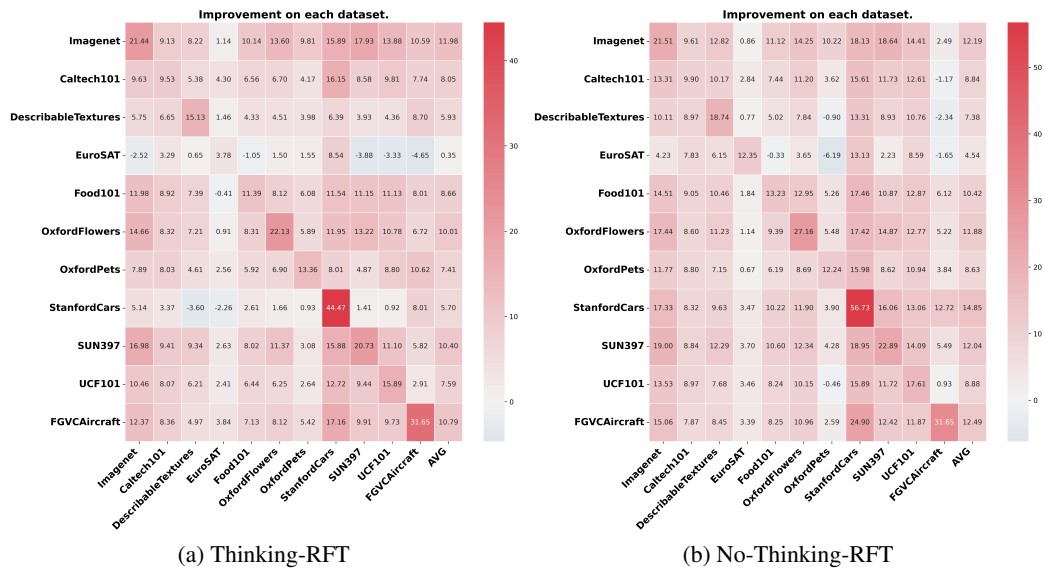

(a) Thinking-RFT            (b) No-Thinking-RFT

Figure 4: Free-lunch phenomenon: Both Thinking-RFT and No-Thinking-RFT lead to cross-dataset accuracy gains. Red: increase; blue: decrease; x-axis: test dataset; y-axis: fine-tuning dataset.

Table 7: Comparison of Thinking-RFT and No-Thinking-RFT with 2B and 7B models on fewshot classification. DTD: DescribableTextures.

| Model | Method | DTD | EuroSAT | OxfordFlowers | StanfordCars | *Average* |
|---|---|---|---|---|---|---|
| 7B | Thinking-RFT | 77.90 | 53.17 | 93.91 | 84.19 | 77.29 |
| | Think-After-Answer | 76.29 | **62.95** | 94.84 | 89.32 | 80.85 |
| | Adaptive-Thinking | 79.60 | 56.62 | **96.86** | 89.92 | 80.75 |
| | No-Thinking-RFT | **80.56** | 58.91 | **94.24** | 94.02 | **81.93** |

Thinking-RFT and No-Thinking-RFT compared with zero-shot Qwen2VL-instruct-2B in Figure 4. We find that when fine-tuned on a specific dataset, both Thinking-RFT and No-Thinking-RFT yield improvements on other datasets in most instances, despite variations in data distribution and even completely different class lists. These results indicate that the application of rule-based verifiable signals and reward loss for model fine-tuning can effectively compel models to acquire essential classification knowledge instead of memorizing. This equips them to achieve superior performance on entirely new datasets. We discuss more about the free lunch phenomenon in Appendix F.4.

## 4.4 Efficiency Comparison

In this subsection, we compare the training and inference efficiency of SFT, Thinking-RFT and No-Thinking-RFT, using the ImageNet dataset as a case study. The results are presented in Table 6. We find that Thinking-RFT requires significantly more time for both training and inference compared to SFT and No-Thinking-RFT, attributable to the necessity of generating multiple lengthy responses during fine-tuning and long reasoning response before answers during inference. In contrast, SFT optimizes only the label tokens during fine-tuning, and No-Thinking-RFT

Table 6: Thinking-RFT requires much more time (in mins) for finetuning and inference *v.s.* SFT and No-Thinking-RFT.

| Method | FT | Infer |
|---|---|---|
| SFT | 35 | 20 |
| Thinking-RFT | 1587 | 30 |
| No-Thinking-RFT | 94 | 26 |

compels the model to output only the ground truth labels during this phase, which significantly reduces the time required. During the inference phase, both methods are designed to output solely class labels, resulting in considerably reduced inference time.

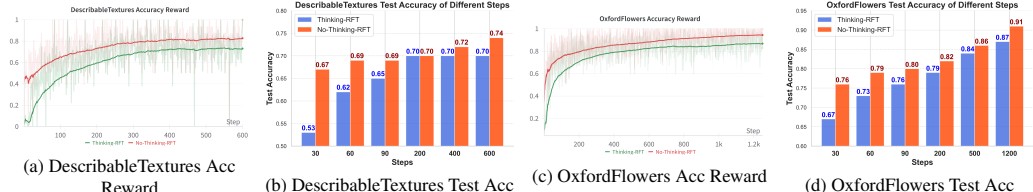

|  | (a) DescribableTextures Acc Reward | (b) DescribableTextures Test Acc | (c) OxfordFlowers Acc Reward | (d) OxfordFlowers Test Acc |

Figure 5: Comparison of accuracy reward convergence speed and test accuracy over steps between Thinking-RFT and No-Thinking-RFT. No-Thinking-RFT converges faster than Thinking-RFT.

## 4.5 Convergence Comparison

In this subsection, we discuss the comparison in convergence rates between Thinking-RFT and No-Thinking-RFT. We illustrate the accuracy reward at each training step and examine the test accuracy at select intervals on the DescribableTextures and OxfordFlowers datasets. The results are shown in Figure 5. We observe that No-Thinking-RFT exhibits a faster convergence speed compared to Thinking-RFT, as evidenced by a higher accuracy reward and significantly higher test accuracy in the early stages of training (within the first 30 steps). These findings imply that slower convergence speed and lower accuracy reward of Thinking-RFT leads to inferior performance than No-Thinking-RFT.

## 5 Related Works

Rule-based RFT has recently achieved significant progress in LLMs [19, 26, 68] and show superior performance compared with SFT. To extend this success to MLLMs, numerous studies have been proposed [41, 60, 82, 24, 46, 3], aiming to replicate phenomena observed in Deepseek-R1, such as increasing response length and emergence of 'aha' moments. Although these initial efforts offer encouraging outcomes, none have thoroughly investigated the role of thinking processes during RFT. Recent research into LLM and MLLM inference suggests that overthinking can degrade performance [27, 64], providing valuable insights into model reasoning. However, these studies exclusively examine the inference stage and do not explore the impact of thinking during the fine-tuning process. As a result, the role of the explicit thinking process in RFT remains unclear. In this paper, we study the effect of explicit thinking during RFT and conduct extensive experiments across different model sizes and tasks. More detailed related works are provided in Appendix B.

## 6 Conclusion

In this paper, we conduct a comprehensive study on the explicit thinking in RFT. We begin by extending Thinking-RFT to classification. Experiments show that Thinking-RFT performs much better than SFT on both base-to-new and few-shot settings. Furthermore, we observed a free-lunch phenomenon with classification RFT, wherein the performance of a model fine-tuned on one dataset improves on other datasets. We then delve into the thinking process of RFT. Inspired by recent research on inference overthinking and observations regarding the decreasing response length of classification RFT, we introduce No-Thinking-RFT, which compels the model to bypass thinking. We conduct extensive experiments on various visual reasoning tasks across different model sizes to evaluate Thinking and No-Thinking-RFT. The results reveal four key findings about the inconsistency in model responses, the impact and the necessity of thinking across different tasks and model sizes, and the stem of No-Thinking-RFT's performance gain. Finally, we propose Thinking-After-Answer to verify the hypothesis that explicit thinking before verifiable reward may hinder fine-tuning and Adaptive-Thinking to explore whether MLLMs can learn when to think or not during RFT.

**Limitations** This study has two main limitations. First, computational constraints restricted our experiments to models 7B and smaller; thus, the applicability of our findings to larger models remains unverified. Second, our Adaptive-Thinking method converges to a single response mode (thinking or No-Thinking) at the task level, rather than adapting at the more granular problem level. Designing methods to achieve this problem-level adaptation is a direction for future work.

## Acknowledgments

This paper is supported by the National Key R&D Program of China No.2022ZD0160101.

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

# Contents

Table 8: Statistics and questions for each Dataset.

| Dataset | Classes | Train | Val | Test | Question |
|---|---|---|---|---|---|
| ImageNet | 1,000 | 1.28M | N/A | 50,000 | "What type of object is in the photo?" |
| Caltech101 | 100 | 4,128 | 1,649 | 2,465 | "What type of object is in the photo?" |
| OxfordPets | 37 | 2,944 | 736 | 3,669 | "What type of object is in the photo?" |
| StanfordCars | 196 | 6,509 | 1,635 | 8,041 | "What type of object is in the photo?" |
| Flowers | 102 | 4,093 | 1,633 | 2,463 | "What type of object is in the photo?" |
| Food101 | 101 | 50,500 | 20,200 | 30,300 | "What type of object is in the photo?" |
| FGVCAircraft | 100 | 3,334 | 3,333 | 3,333 | "What type of aircraft is in the photo?" |
| SUN397 | 397 | 15,880 | 3,970 | 19,850 | "What type of object is in the photo?" |
| DTD | 47 | 2,820 | 1,128 | 1,692 | "What type of texture is in the photo?" |
| EuroSAT | 10 | 13,500 | 5,400 | 8,100 | "What type of object is in the centered satellite photo?" |
| UCF101 | 101 | 7,639 | 1,898 | 3,783 | "What actions is the person performing in the photo?" |

# A    Group Relative Policy Optimization

We follow Deepseek-R1 [19, 59] to employ Group Relative Policy Optimization (GRPO) as the RL algorithm for optimization. Unlike SFT, which optimizes models through token-level losses, RL-based methods like GRPO utilize policy gradients, calculated from reward loss, for optimization. This encourages reasoning by exploring a much larger solution space [19].

Let $Q$ be the qestion set, $\pi_{\theta_{\text{old}}}$ be the policy model and $\{o_1, o_2, \cdots, o_G\}$ be a group of response from $\pi_{\theta_{\text{old}}}$ for a question $q$. Let $\pi_{\theta_{\text{ref}}}$ denote the frozen reference model. The GRPO algorithms aim to optimize model $\pi_\theta$ by the following objective:

$$J_{\text{GRPO}}(\theta) = \mathbb{E}_{q \sim Q, \{o_i\}_{i=1}^G \sim \pi_{\theta_{\text{old}}}}$$

$$\left[ \frac{1}{G} \sum_{i=1}^G \min \left( \frac{\pi_\theta(o_i|q)}{\pi_{\theta_{\text{old}}}(o_i|q)} A_i, \text{clip} \left( \frac{\pi_\theta(o_i|q)}{\pi_{\theta_{\text{old}}}(o_i|q)}, 1 - \epsilon, 1 + \epsilon \right) A_i \right) - \beta D_{\text{KL}}(\pi_\theta \| \pi_{\text{ref}}) \right],$$

where $\epsilon$ and $\beta$ are clipping hyper-parameter and the coefficient controlling the Kullback–Leibler (KL) penalty, respectively. Here, $A_i = \frac{r_i - \text{mean}(\{r_1, r_2, \ldots, r_G\})}{\text{std}(\{r_1, r_2, \ldots, r_G\})}$ is the advantage using the group reward $\{r_1, r_2, \ldots, r_G\}$, and $D_{KL}(\pi_\theta \| \pi_{\text{ref}}) = \frac{\pi_{\text{ref}}(o_i|q)}{\pi_\theta(o_i|q)} - \log \left( \frac{\pi_{\text{ref}}(o_i|q)}{\pi_\theta(o_i|q)} \right) - 1$ is the KL divergence loss to prevent current model $\pi_\theta$ deviating too much from reference model $\pi_{\theta_{\text{ref}}}$. GRPO eliminates the critic model in PPO by estimating the relative advantage by sampling a group of responses $\{o_i\}_{i=1}^G$ and normalizing their rewards within the group to compute a relative advantage, which is more computationally efficient [59].

# B    Detailed Related Works

## B.1    MLLM for Image Classification

Image classification is at the core of machine learning research, driving many fundamental advancements in theoretical understanding and practical applications. Early CNN-based [35, 32, 62, 66, 20, 37, 22, 67] models perform classification by hierarchically extracting spatial features through convolutional layers and mapping them to class probabilities. Vision transformers [15] largely follow the same pipeline but replace convolution with self-attention [70] for global feature modeling. CLIP [55] classifies images by computing the similarity between the image embedding and text embeddings of category descriptions and then selecting the closest match. More recently, the advent of MLLM enabled a new paradigm for image classification that leverages both vision encoders and LLMs. Unlike traditional classifiers, MLLMs take images and natural language prompts as input and generate text-based outputs, making them more interpretable, flexible, and user-friendly. However, MLLM for image classification is still an underexplored area. Recent work [81] shows that MLLMs perform poorly at classification but can be improved with SFT. In this work, we take a different angle and investigate a novel approach: using rule-based RL to fine-tune MLLM for classification. We systematically study its effectiveness compared to SFT and show its advantage and potential improvements.

## B.2 RL for Post Training

Reinforcement learning (RL) has become an important technique used in LLM and MLLM post-training. Introduced by InstructGPT [50] as Reinforcement Learning from Human Feedback (RLHF), RL was used to align LLM and MLLM output with human's preference. Due to the computation inefficiency of PPO [58] used in RLHF, some offline RL algorithms [47, 56] and value model free RL algorithms [59, 23] were introduced in the post training stage. Besides, researchers also explored how to build a good reward model in terms of robustness and reward density [73, 18, 39]. Recently, DeepSeek-R1 [19] applied the rule-based reward to the reinforcement training of LLM, proving the huge potential of RL in terms of incentivizing LLM's reasoning ability. Inspired by DeepSeek-R1, we applied reinforcement learning to the classification task with a rule-based reward function, and analyzed its generalizability and learning efficiency compared to supervised fine-tuning.

## B.3 Rule-Based Reinforcement Fine-tuning

Rule-based reinforcement fine-tuning (RFT) has recently achieved much process in large language models [19, 26, 68]. To transfer this success to MLLMs, numerous studies have been proposed [41, 60, 82, 24, 46, 3, 34, 53]. These works aim to replicate phenomena observed in Deepseek-R1, such as increased response length and the emergence of 'aha' moments. However, the role of the thinking process in RFT has not been explored in depth in these works. In this paper, we investigate its impact across different model sizes and tasks.

## B.4 Overthinking in LLMs and MLLMs

Recent advancements in sophisticated reasoning abilities enabled by techniques such as Chain-of-Thought (CoT) prompting [74] have marked a significant milestone in the development of large language models (LLMs) and multimodal LLMs (MLLMs). CoT allows models to generate intermediate reasoning steps when solving complex problems, thereby improving both transparency and performance. However, this capability has also introduced a notable challenge referred to as the "overthinking phenomenon"[65].

Overthinking describes the tendency of LLMs and MLLMs to produce unnecessarily verbose, redundant, and computationally expensive reasoning chains, even for simple queries. This behavior can hinder practical deployment and, in some cases, degrade performance [40, 4, 10, 75, 28, 27, 29, 78, 11]. For example, [40] shows that CoT can harm accuracy on tasks where extra deliberation impairs human performance. [4] quantifies overthinking in powerful LLMs and proposes pruning strategies, [10] shows excessive internal reasoning degrades success of LLM-based agents and [27] demonstrates that reasoning offers limited benefits on commonsense tasks. Additionally, [28] shows that overthinking can negatively impact MLLM inference accuracy and [78] enhances CoT reasoning by representing its steps in a continuous space which makes CoT more efficient and improve performance.

While these studies provide valuable insights into model reasoning behavior, they focus exclusively on inference. The effect of explicit reasoning during reinforcement fine-tuning (RFT), however, remains largely unexplored.

## C  Dataset Statistics and Details

Following CoOp [84], we conducted extensive experiments on 11 public classification benchmark datasets to evaluate the effectiveness of the proposed CLIPFit. The datasets are ImageNet [14], Caltech101 [16], OxfordPets [51], StanfordCars [31], Flowers102 [48], Food101 [2], FGVCAircraft [44], SUN397 [76], DTD [9], EuroSAT [21], and UCF101 [63]. The dataset statistics and the questions for each dataset are shown in Table 8.

For general reasoning dataset, CVBench is a benchmark for spatial reasoning and SAT is a training dataset for spatial reasoning. PuzzleVQA is a dataset for visual puzzle reasoning. MathVista is for general math reasoning and MathVision is for competition math reasoning.

The examples of these datasets can be found in Appendix I.

Table 9: Comparison of Qwen2VL instruct, SFT, Thinking-RFT, and No-Thinking-RFT in the base-to-new generalization setting. No-Thinking: No-Thinking-RFT. Base: base class accuracy. New: new class accuracy. H: harmonic mean accuracy. No-Thinking: No-Thinking-RFT.

(a) Average over 11 datasets.

|  | Base | New | H |
|---|---|---|---|
| Qwen2VL | 62.1 | 66.27 | 64.12 |
| SFT | 67.4 | 70.73 | 69.03 |
| Thinking-RFT | 81.17 | 79.15 | 80.15 |
| No-Thinking | **83.42** | **81.88** | **82.64** |

(b) ImageNet.

|  | Base | New | H |
|---|---|---|---|
| Qwen2VL | 61.56 | 74.9 | 67.58 |
| SFT | 27.78 | 47.78 | 35.13 |
| Thinking-RFT | 88.12 | 90.01 | 89.05 |
| No-Thinking | **88.97** | **90.66** | **89.81** |

(c) Caltech101.

|  | Base | New | H |
|---|---|---|---|
| Qwen2VL | 88.83 | 92.9 | 90.82 |
| SFT | 93.87 | 93.01 | 93.44 |
| Thinking-RFT | 97.74 | 95.2 | 96.45 |
| No-Thinking | **97.93** | **95.63** | **96.77** |

(d) DescribableTextures.

|  | Base | New | H |
|---|---|---|---|
| Qwen2VL | 60.99 | 61.34 | 61.17 |
| SFT | 71.98 | 71.41 | 71.7 |
| Thinking-RFT | 77.42 | 67.82 | 72.3 |
| No-Thinking | **77.42** | **70.37** | **73.72** |

(e) EuroSAT.

|  | Base | New | H |
|---|---|---|---|
| Qwen2VL | 54.52 | 63.54 | 58.69 |
| SFT | **91.55** | **77.87** | **84.16** |
| Thinking-RFT | 58.09 | 69.33 | 63.22 |
| No-Thinking | 66.43 | 74.13 | 70.07 |

(f) Food101.

|  | Base | New | H |
|---|---|---|---|
| Qwen2VL | 74.07 | 80.42 | 77.12 |
| SFT | 74.27 | 77.82 | 76.0 |
| Thinking-RFT | 87.29 | 87.56 | 87.42 |
| No-Thinking | **88.77** | **88.93** | **88.85** |

(g) OxfordFlowers.

|  | Base | New | H |
|---|---|---|---|
| Qwen2VL | 61.1 | 60.99 | 61.05 |
| SFT | **97.77** | **94.95** | **96.34** |
| Thinking-RFT | 87.78 | 74.97 | 80.87 |
| No-Thinking | 88.71 | 76.73 | 82.29 |

(h) OxfordPets.

|  | Base | New | H |
|---|---|---|---|
| Qwen2VL | 75.59 | 91.79 | 82.9 |
| SFT | 84.06 | 86.28 | 85.15 |
| Thinking-RFT | 83.28 | 94.49 | 88.53 |
| No-Thinking | **86.64** | **95.5** | **90.85** |

(i) StanfordCars.

|  | Base | New | H |
|---|---|---|---|
| Qwen2VL | 43.81 | 33.15 | 37.74 |
| SFT | 74.54 | 69.68 | 72.03 |
| Thinking-RFT | 82.08 | 75.74 | 78.78 |
| No-Thinking | **91.13** | **87.04** | **89.04** |

(j) SUN397.

|  | Base | New | H |
|---|---|---|---|
| Qwen2VL | 56.97 | 65.8 | 61.07 |
| SFT | 27.39 | 37.8 | 31.77 |
| Thinking-RFT | 81.03 | 82.52 | 81.77 |
| No-Thinking | **83.18** | **84.14** | **83.66** |

(k) UCF101.

|  | Base | New | H |
|---|---|---|---|
| Qwen2VL | 69.6 | 64.62 | 67.02 |
| SFT | 59.95 | 63.93 | 61.87 |
| Thinking-RFT | 79.47 | 74.95 | 77.14 |
| No-Thinking | **80.47** | **79.18** | **79.82** |

(l) FGVCAircraft.

|  | Base | New | H |
|---|---|---|---|
| Qwen2VL | 36.07 | 39.47 | 37.7 |
| SFT | 38.23 | 57.53 | 45.94 |
| Thinking-RFT | **70.53** | 58.07 | **63.69** |
| No-Thinking | 68.01 | **58.31** | 62.79 |

# D   Implementation Details

## D.1   Dataset Construction for Classification

Due to computational resource and model input limit, we select subset of class labels as the choice list. For few-shot classification, we sample 40% of labels including ground truth labels to form the classification list in question. For base-to-new classification, we sample 80% of labels including ground truth labels to form the classification list in question. If the length of the final choice list is less than 30, we include all class labels in the choice list, and if the length of the final choice list is larger than 100, we include 100 class labels in the choice list.

## D.2   Implementation Details for Classification

We implement our code in Pytorch [52]. We utilize Qwen2-VL-2B-Instruct [72] as the base model, and fine-tune all parameters during training, following [82, 3]. All training is conducted in 8 A100 GPUs. The batch size is set to 1 per GPU and we use 2-step gradient accumulation during training. All images are resized to 328×328 resolution with no data augmentation. We first extract answers from answer tags (<answer> ...</answer>) and then verify if class names are in answers. If the answer tag does not exist in model responses we directly verify if class names are in model responses, following [81]. Both the maximum prompt length and maximum response length are set to 1024 for all datasets except StanfordCars dataset. The maximum prompt length and maximum response length are set to 1024 for StanfordCars dataset are set to 2048 and 1024, respectively. The number of rollout is set to 4 for both 2B and 7B models. $\beta$ is set to 0.04, learning rate is set to $1 \times 10^{-6}$.

### D.3 Implementation Details for More Diverse Tasks

**Visual Perception.** We follow [82] to fine-tune models on SAT dataset [57] 2 epochs and then test on CVBench dataset [69]. We also include classification results for comparison. The maximum length of prompt and response is both set to 1024. The number of rollout is set to 4 for both 2B and 7B models. The number of rollout is set to 4 for both 2B and 7B models. $\beta$ is set to 0.04, learning rate is set to $1 \times 10^{-6}$. The batch size is set to 1 per GPU and we use 2-step gradient accumulation during training.

**Multi-Modal Math Reasoning.** We utilize the Math-40K [61] as the find-tuning data and fine-tune models 1 epoch. Then we test the fine-tuned models on both MathVista [43] and MathVision [71]. The maximum length of input prompt and response are set to 4096 and 512 respectively. The number of rollout is set to 8 for 2B model, and 4 for 7B model. $\beta$ is set to 0.04, learning rate is set to $1 \times 10^{-6}$. The batch size is set to 1 per GPU and we use 2-step gradient accumulation during training.

**Visual Puzzle Reasoning.** We follow the code of [7] to generate a training dataset with 6.5k data and fine-tune models 2 epochs. We then test fine-tuned models on PuzzleVQA [7] as in-domain testing, and on AlgoPuzzleVQA [17] as out-of-domain (OOD) testing. $\beta$ is set to 0.04, learning rate is set to $1 \times 10^{-6}$. The batch size is set to 1 per GPU and we use 2-step gradient accumulation during training. The maximum length of prompt and response is both set to 1024. The number of rollout is set to 8 for 2B model, and 4 for 7B model.

# E Prompt

## E.1 Few-shot Prompt

We provide the prompt for thinking answer extract here. We use classification as the example, and for other datasets, the only difference is the few-shot examples.

**Prompt:** *I will give you a question, answer and the model response. The model response is in the form <thinking>...</thinking><answer>...</answer>. The thinking process is in <thinking>...</thinking> and the answer is in <answer>...</answer>.*
*I want to to extract the answer in thinking tags and see if it is the same as the answer in answer tag. For multichoice problem, extract the choice (e.g., A,B,C) that corresponding to the answer meaning of thinking process in thinking tag. If there is no answer in thinking tag, output None. Please output three items answer in thinking tag: answer in answer tag:*
*consistency with answer tag:*
*Below is some examples:*
*Question: What type of texture is in the photo?*
*Please choose one from list [ perforated, marbled, zigzagged, waffled, meshed, flecked, sprinkled, matted, woven, grooved, gauzy, bumpy, polka-dotted, potholed, veined, frilly, grid, bubbly, cobwebbed, scaly, pleated, crosshatched, wrinkled, freckled, porous, stained, swirly, chequered, spiralled, fibrous].*
*answer: veined*
*model response: <think> The texture in the photo appears to be a series of parallel lines or grooves running along the surface of the leaf. This suggests a grooved texture. </think> <answer> grooved </answer>*
*answer in thinking tag: grooved*
*answer in answer tag: grooved*
*consistency with answer tag: Yes*
*Question: What type of texture is in the photo?*
*Please choose one from list [ fibrous, striped, chequered, swirly, studded, lacelike, knitted, freckled, crosshatched, wrinkled, cobwebbed, grooved, waffled, grid, scaly, bubbly, sprinkled, paisley, spiralled, interlaced, porous, veined, blotchy, pitted, zigzagged, crystalline, flecked, potholed, cracked, honeycombed].*
*answer: potholed*
*model response:<think> The photo shows a road with visible damage, including cracks and potholes. The texture appears to be characterized by these irregularities and depressions in the surface. </think> <answer> potholed </answer>*
*answer in thinking tag: None*
*answer in answer tag: potholed*

Table 10: Comparison of Adaptive-Thinking model response type among 2B and 7B models on MathVista, MathVision, CVBench, PuzzleVQA (Puzzle), and AlgoPuzzleVQA (AlgoPuzzle).

| Model | Response Form | MathVista | MathVision | CVBench | Puzzle | AlgoPuzzle |
|-------|---------------|-----------|------------|---------|--------|------------|
| 2B | w/ Thinking | 0% | 0% | 0% | 0% | 0% |
|    | w/o Thinking | 100% | 100% | 100% | 100% | 100% |
| 7B | w/ Thinking | 100% | 100% | 99.6% | 0% | 0% |
|    | w/o Thinking | 0% | 0% | 0.4% | 100% | 100% |

*consistency with answer tag: None*
*Question: What type of texture is in the photo?*
*Please choose one from list [ dotted, knitted, woven, honeycombed, bubbly, lacelike, swirly, interlaced, spiralled, marbled, zigzagged, pleated, stratified, gauzy, chequered, fibrous, paisley, lined, striped, studded, grid, crosshatched, crystalline, blotchy, stained, wrinkled, frilly, scaly, waffled, perforated].*
*answer: spiral*
*model response: <think> The photo shows a pattern with concentric circles that twist and turn in a spiral-like manner. This suggests that the texture is related to the spiral pattern.</think> <answer> swirly </answer>*
*answer in thinking tag: spiral*
*answer in answer tag: swirly*
*consistency with answer tag: No*
*(two more examples)*

## E.2   Thinking-After-Answer

The prompt of Think-After-Answer is designed as: *{Question} Please first output the answer in <answer> </answer> tags and then output a brief reasoning process in <reason> </reason> tags.*

## E.3   Adaptive-Thinking

We provide more discussions about Adaptive-Thinking method in this subsection.

**Instruction Prompt.** *{Question}. Please first identify whether this problem requires intermediate thinking or calculation. If the problem requires thinking or calculation, output the thinking and calculation process inside <think> </think> tags and the final answer inside <answer> </answer> tags. If no thinking or calculation is required, directly output the final answer inside <answer> </answer> tags. Your output should follow one of two cases: (1) '<answer> ... </answer>', (2) '<think> ... </think> <answer> ... </answer>'.*

**Format Reward.** Both the thinking format (<think> ... </think> <answer> ... </answer>) and the direct-answer format (<answer> ... </answer>) will be assigned a reward of 1.

**Accuracy Reward.** The accuracy reward is not changed as Thinking-RFT.

# F   More Experimental Results on Classification

## F.1   Base-to-New Results

In this subsection, we introduce the results on the base-to-new generalization setting. Following [84], We split each dateset into two disjoint groups: the base class dataset and the new class dataset. This experimental setup is designed to assess the model's ability to acquire downstream knowledge while simultaneously demonstrating generalization to novel information. All the methods are fine-tuned on the base class dataset and evaluated on both the base and new class test dataset. We conduct 4-shot experiments. The results are shown in Table 9.

As demonstrated in Table 9, our proposed Thinking-RFT framework notably surpasses SFT in performance. In detail, Thinking-RFT exhibits an approximately 14% higher accuracy for base classes and a 9% increase for new class accuracy, leading to an aggregate improvement of 11% in the harmonic mean accuracy. These results demonstrate the effectiveness of rule-base reinforcemnet

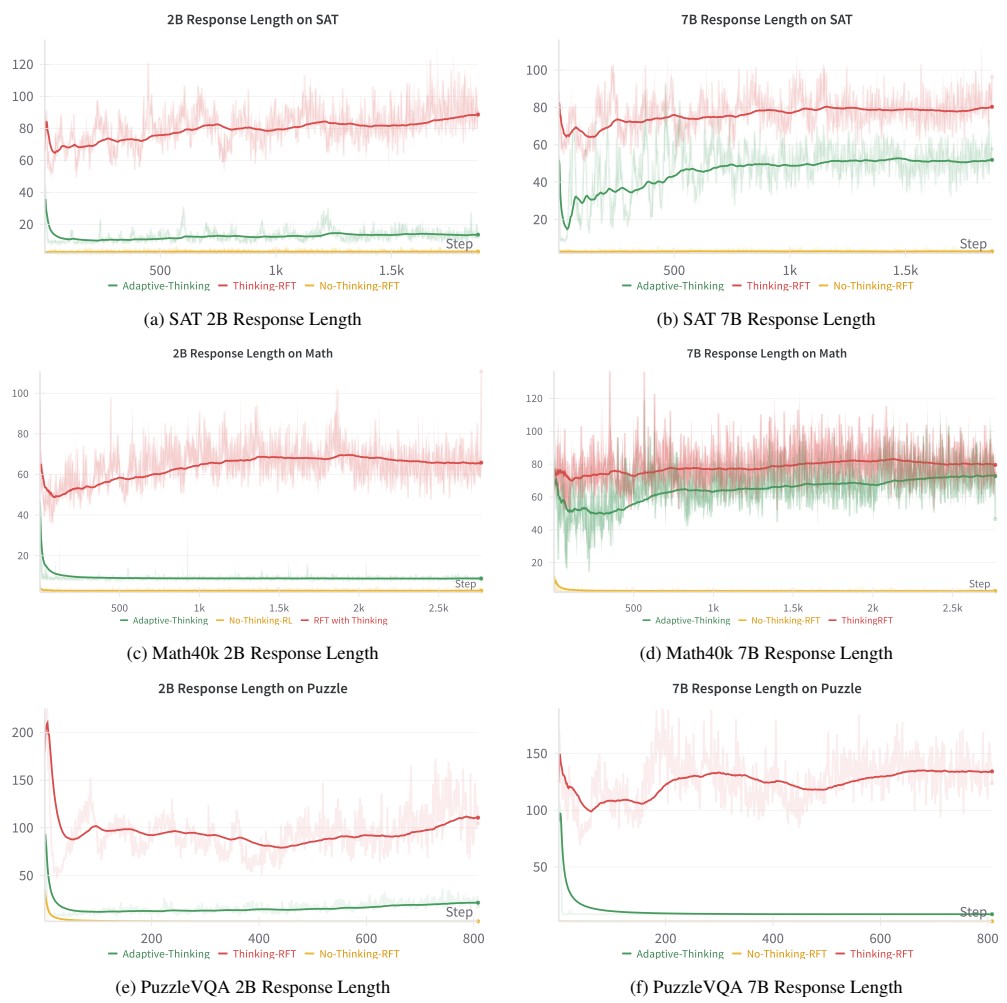

Figure 6: Comparison of response length trend on SAT, Math40k, and PuzzleVQA datasets over steps of Thinking-RFT, No-Thinking-RFT , and Adaptive-Thinking across 2B and 7B models.

fine-tuning in image classification. However, we also find that SFT can surpass Thinking-RFT in certain datasets, such as OxfordFlowers and EuroSAT, indicating that SFT may hold advantages in specific scenarios. We also observed that SFT significantly underperforms on the ImageNet and SUN397 datasets. The likely reason is that the prompts in these two datasets are very lengthy, preventing SFT from effectively memorizing classification knowledge, thereby resulting in poor performance.

The proposed No-Thinking-RFT demonstrates superior performance, in both base class and new class average accuracy, resulting in a 2.5% enhancement in average harmonic mean accuracy, compared with Thinking-RFT. These findings suggest that omitting the thinking process during fine-tuning allows rule-based RL to achieve improved classification performance and enhanced generalization capabilities than with thinking process.

## F.2 Open-set Classification Comparison

We present the results of open-set classification using Thinking-RFT and No-Thinking-RFT in this subsection. Unlike closed-form classification, open-set classification is not a well-defined problem, which is much harder or even unrealistic for some datasets, since synonyms, plural forms, and name partially missing will be judged as incorrect. For example, in the StanfordCars dataset [31], the model can hardly output the correct year of the car in images. Therefore, we selected five datasets with class names that are relatively straightforward for the model to output and compare the few-shot

Table 11: Comparison on open-set Few-shot learning results.

| | ImageNet | Caltech101 | Food101 | Flowers102 | OxfordPets | *Average* |
|---|---|---|---|---|---|---|
| Qwen2VL | 46.57 | 62.96 | 57.79 | 48.44 | 47.40 | 52.63 |
| Thinking-RFT | 54.84 | 79.07 | **73.51** | 67.64 | **89.94** | 73.0 |
| No-Thinking-RFT | **56.45** | **86.29** | 71.99 | **71.21** | 86.07 | **74.40** |

Table 12: Comparison of Thinking-RFT and No-Thinking-RFT with 2B and 7B models on fewshot classification. DTD: DescribableTextures.

| Model | Method | DTD | EuroSAT | OxfordFlowers | StanfordCars | *Average* |
|---|---|---|---|---|---|---|
| 2B | Thinking-RFT | 69.92 | 49.46 | 86.56 | 80.24 | 71.55 |
| | No-Thinking-RFT | **73.52** | **58.02** | **91.6** | **92.5** | **78.91** |
| 7B | Thinking-RFT | 77.90 | 53.17 | 93.91 | 84.19 | 77.29 |
| | Think-After-Answer | 76.29 | **62.95** | 94.84 | 89.32 | 80.85 |
| | Adaptive-Thinking | 79.60 | 56.62 | **96.86** | 89.92 | 80.75 |
| | No-Thinking-RFT | **80.56** | 58.91 | **94.24** | 94.02 | **81.93** |

learning performance between Thinking-RFT and No-Thinking-RFT. The results are shown in Table 11. No-Thinking-RFT outperforms Thinking-RFT on three datasets among five datasets, ultimately achieving a 1.4% improvement in average accuracy over Thinking-RFT.

### F.3 Results of Classification with 7B Models.

We further conduct experiments of classification on DescribableTextures, EuroSAT, OxfordFlowers, StanfordCars datasets using Qwen-VL-2-7B models on and report the results on Table 12. As shown in Table 12, compared with 2B models, the performance gap between Thinking-RFT and No-Thinking-RFT narrows but remains. These results suggest that image classification does not require explicit thinking, and that excluding the explicit thinking process can enhance both performance and computational efficiency.

### F.4 More Discussion About Free-Lunch Phenomenon

In this subsection, we give more discussion about the free-lunch phenomenon. Despite improvement in most cases for cross dataset improvement in Figure 4, it is also noted that such improvements can be negative in certain instances. For example, fine-tuning on the EuroSAT dataset could result in diminished performance on the OxfordPets dataset. This outcome is understandable, given that the knowledge required for the EuroSAT dataset is significantly divergent from that of the OxfordPets dataset, and the classification knowledge from the EuroSAT dataset may even be detrimental to the classification of the OxfordPets dataset.

We further illustrate the comparative improvement of No-Thinking-RFT over Thinking-RFT by testing on 11 datasets, using a model that was fine-tuned on one specific dataset. We also showcase the comparative improvement achieved by using a standard training prompt over a No-Thinking Prompt (which directs Thinking-RFT to immediately produce the answer during inference). The results are shown in Figure 7. We can find that No-Thinking-RFT has a better cross-dataset generalization ability than Thinking-RFT, except OxfordPets and FGVCAircraft datasets. This suggests that utilizing an equality reward without engaging in any thinking process can enhance the model's cross-dataset generalization ability, with the exceptions being the OxfordPets and FGVCAircraft datasets. For these datasets, the thinking process during fine-tuning appears to be important for cross-dataset performance. These findings are consistent to the results of using different inference prompt strategies of Thinking-RFT, as shown in Figure 7 (b). We find that using normal prompt for in-domain fewshot test performs better than No-Thinking prompt among all datasets. This is reasonable given that the normal prompt is the one used for fine-tuning. However, it is surprising to discover that using the normal prompt for cross-dataset testing results in lower performance in many cases compared to the No-Thinking prompt, except OxfordPets and FGVCAircraft datasets. These findings suggest

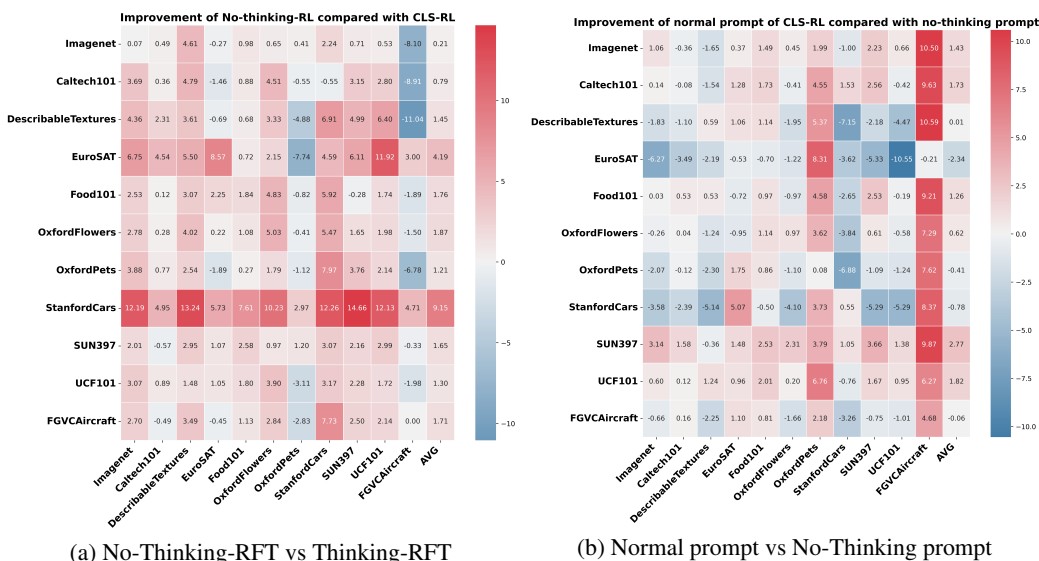

(a) No-Thinking-RFT vs Thinking-RFT

(b) Normal prompt vs No-Thinking prompt

Figure 7: Visualization of improvement of No-Thinking-RFT on different datstes compared with Thinking-RFT (Left) and improvement of Thinking-RFT compared with Thinking-RFT with No-Thinking prompt (Right). The accuracy improvement is marked as red, and decrease is marked as blue

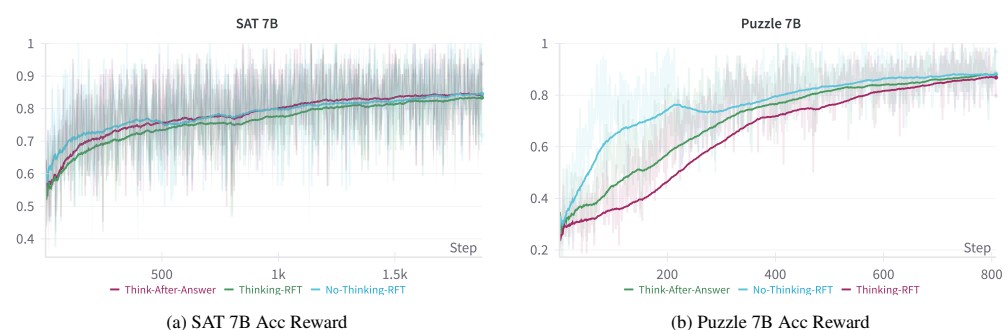

(a) SAT 7B Acc Reward

(b) Puzzle 7B Acc Reward

Figure 8: Comparison of accuracy reward convergence speed on Puzzle dataset over steps of Thinking-RFT, Think-After-Answer, adn No-Thinking-RFT across 2B and 7B models.

that while Thinking-RFT fine-tuning may enable the model to learn good cross-dataset classification generalization ability, over-thinking during inference can potentially diminish this ability.

## G More Experimental Results on More Diverse Tasks and Model Types

### G.1 Experiments on More different Models

We report more experimental results on more diverse model types. We conduct experiments on InternVL2.5-1B and InternVL2.5-4B [5], and Qwen2-5-VL-Instruct-3B and Qwen2-5-VL-Instruct-7B [1]. The experimental results are shown in Table 14 and Table 13. The results are almost similar to which in the main text, i.e., visual and puzzle tasks do not need thinking during RFT.

### G.2 Referring Expression Comprehension Results

We follow VLM-R1 [60] to conduct experiments of the referring expression comprehension task. The experiment setting and implementation is the same as VLM-R1 [60]. We follow [60] to use the training splits of Refcoco/+/g [79, 45] as the training data, and the val split of Refcoco/+/g for in-domain evaluation and test split of LISA-Grounding [33] for out-of-domain (OOD) evaluation.

Table 13: Results of Thinking-RFT and No-Thinking-RFT on CVBench across different models.

| Model | Setting | Overall | Count | Relation | Depth | Distance |
|-------|---------|---------|-------|----------|-------|----------|
| InternVL-2.5-1B | Thinking-RFT | 68.57 | 62.82 | **80.46** | 75.33 | 56.50 |
|  | No-Thinking-RFT | **70.55** | **65.48** | 77.08 | **78.67** | **62.00** |
| InternVL-2.5-4B | Thinking-RFT | 79.34 | **71.32** | **91.08** | **87.17** | 69.33 |
|  | No-Thinking-RFT | **79.76** | 70.94 | 89.85 | 86.50 | **73.67** |
| Qwen-VL-2.5-3B | Thinking-RFT | 76.76 | **70.43** | 86.46 | 79.67 | 71.67 |
|  | No-Thinking-RFT | **79.83** | 70.18 | **89.54** | **83.67** | **78.17** |

Table 14: Performance comparison of PuzzleVQA and AlgoPuzzleVQA on Qwen2-5-VL-Instruct-7B models under three settings.

| Model | Method | PuzzleVQA | AlgoPuzzleVQ |
|-------|--------|-----------|--------------|
| Qwen-VL-2.5-7B | Thinking-RFT | 77.10 | **27.72** |
|  | No-Thinking-RFT | **86.45** | 26.11 |

We follow VLM-R1 [60] to report the evaluation results of first 600 steps. The results are shown in Table 15. We observe that No-Thinking-RFT generally achieves better performance than VLM-R1 on the RefCOCO/+/g in-domain tests but performs worse on the LISA-Grounding out-of-domain (OOD) test. These results suggest that for referring grounding tasks, reinforcement fine-tuning (RFT) without explicit thinking can improve in-domain performance while potentially degrading OOD generalization. However, we also note that as fine-tuning progresses, the OOD performance of No-Thinking-RFT improves and consistently surpasses that of SFT.

## G.3 Experiments on Few-Shot Detection

We follow Viusal-RFT [41] to conduct experiments on few-shot object detection with MLLMs. We follow [41] to select eight categories from the COCO dataset with 4 images per category, to construct training dataset. We then follow [41] to train the Qwen2-VL-2B-instruct-2B and Qwen2-VL-2B-instruct-7B models for 100 steps (50 epochs). The results are shown in Table 16. No-Thinking-RFT consistently outperforms Thinking-RFT among both 2B and 7B models.

## G.4 Experiments of Thinking-RFT inference with no-thinking tags.

We further conduct experiments to investigate where the performance gain of No-Thinking-RFT over Thinking-RFT on perception and puzzle tasks originates. We explore the No-Thinking inference mode of Thinking-RFT by appending an empty thinking tag (*<thinking> </thinking>*) during inference. The results are shown in Table 17. We find that while appending empty thinking tags during inference could improve performance on CVBench and puzzle tasks, its performance is still far behind No-Thinking-RFT. Moreover, appending empty thinking tags during inference will decrease performance on math tasks. These results suggest that the performance gain of No-Thinking-RFT over Thinking-RFT on perception and puzzle tasks primarily stems from two factors: improved learning during fine-tuning and the avoidance of overthinking by bypassing inference.

# H More Discussion

## H.1 Limited Gains in Puzzle Tasks

In this subsection, we discuss the explanation accounts for limited puzzle gains. As a pragmatic proxy, we evaluated the o4-mini model, whose API supports optional in-context visual reasoning (e.g., cropping). On PuzzleVQA, performance rises from 84.30% (no visual reasoning) to 86.48% (with visual reasoning), a gain of +2.18% accuracy. This suggests that visual-context learning can improve puzzle performance.

Table 15: Performance comparison of Refcoco/+/g and LISA ground datasets.

| Training method | Evaluation Dataset | 100 | 200 | 300 | 400 | 500 | 600 |
|---|---|---|---|---|---|---|---|
| SFT | Refcoco$_{val}$ | 88.7 | 88.85 | 88.7 | 88.25 | 88.85 | 88.7 |
| VLM-R1 | Refcoco$_{val}$ | 88.7 | 88.7 | 89.4 | 89.25 | 90 | 90.55 |
| No-Thinking-RFT | Refcoco$_{val}$ | **90.29** | **90.85** | **90.52** | **90.88** | **90.85** | **90.83** |
| SFT | Refcoco+$_{val}$ | 82.55 | 82.15 | 81.85 | 81.9 | 82.3 | 82.25 |
| VLM-R1 | Refcoco+$_{val}$ | 82.6 | 81.9 | 82.8 | 83.35 | 83.6 | 84.3 |
| No-Thinking-RFT | Refcoco+$_{val}$ | **83.71** | **84.39** | **84.76** | **85.02** | **85.28** | **85.24** |
| SFT | Refcocog$_{val}$ | 85.65 | 85.95 | 85.85 | 85.6 | 85.95 | 85.95 |
| VLM-R1 | Refcocog$_{val}$ | **85.95** | 85.05 | 85.45 | 85.65 | **87.15** | **87.1** |
| No-Thinking-RFT | Refcocog$_{val}$ | 85.70 | **86.15** | **86.70** | **86.60** | 86.97 | 86.91 |
| SFT | LISA-Grounding | 55.91 | 56.51 | 55.66 | 55.18 | 55.66 | 54.82 |
| VLM-R1 | LISA-Grounding | **61.82** | **61.27** | **61.64** | **62.6** | **61.88** | **63.14** |
| No-Thinking-RFT | LISA-Grounding | 57.90 | 57.24 | 58.99 | 59.65 | 59.95 | 61.76 |

Table 16: Performance comparison of 4-shot results on COCO dataset of 8 categories.

| Models | mAP | bus | train | fire hydrant | stop sign | cat | dog | bed | toilet |
|---|---|---|---|---|---|---|---|---|---|
| | | | | *Qwen2-VL-2B* | | | | | |
| Zero-Shot | 19.6 | 19.0 | 15.8 | 25.8 | 18.4 | 29.9 | 23.2 | 14.6 | 9.8 |
| Thinking-RFT | 40.6 | 30.0 | 40.6 | 45.7 | 35.0 | 60.9 | 44.9 | 24.6 | **43.1** |
| No-Thinking-RFT | **43.72** | **32.29** | **44.64** | **47.38** | **43.43** | **61.59** | **52.94** | **24.81** | 42.71 |
| | | | | *Qwen2-VL-7B* | | | | | |
| Zero-shot | 43.0 | 35.0 | 43.3 | 37.1 | 36.7 | 57.3 | 50.3 | 37.4 | 47.1 |
| Thinking-RFT | 54.3 | 44.3 | **59.8** | 52.0 | **46.0** | **72.7** | 62.8 | 41.9 | **55.0** |
| No-Thinking-RFT | **54.99** | **49.10** | 58.72 | **53.37** | 45.52 | 72.35 | **63.64** | **46.41** | 50.79 |

## H.2 Discussion between offline DPO, No-Thinking-RFT, and Thinking-RFT

In this subsection, we discuss the difference between offline DPO, No-Thinking-RFT, and Thinking-RFT.

(i) Although No-Thinking-RFT removes the "think" phase, a list-wise DPO algorithm **should not** be expected to reproduce similar effect due to the fundamental difference between DPO and GRPO. We discuss the detailed difference below:

   (a) **Sampling strategy**. List-wise DPO selects one positive and several negatives *offline*. In contrast, No-Thinking-RFT samples responses *online* at every training step: the mix of positives and negatives is not fixed at $1 : (N - 1)$ but varies with task difficulty and model competence (from $0 : N$ to $N : 0$). Thus while removing thinking, No-Thinking-RFT continues to explore and learn from fresh errors, whereas DPO does not.

   (b) **Gradient update rule**. No-Thinking-RFT computes a policy-gradient using verifiable rewards ($R = 1$ for an exact match, $\mathcal{R} = 0$ otherwise) and a group-advantage formulation, i.e. GRPO. Listwise DPO, by contrast, minimises a cross-entropy loss over a pre-computed list, functioning more like contrastive SFT. This distinction yields different learning dynamics.

   Because of these differences, listwise DPO **should not** be expected to match the empirical behaviour of No-Thinking-RFT.

(ii) We also implemented preliminary DPO and listwise DPO baselines to quantify the performance difference. We evaluated two negative-sampling strategies: one using samples generated by Qwen2-VL (DPO model) and the other using randomly generated samples (DPO random). Experiments were carried out on the FGVCAircraft classification dataset and the MathVQA dataset. The results shown in Table 18 indicate that both DPO variants under both sampling strategies perform substantially worse than Thinking-RFT and

Table 17: Comparison of Adaptive-Thinking model response type among 2B and 7B models on MathVista, MathVision, CVBench, PuzzleVQA (Puzzle), and AlgoPuzzleVQA (AlgoPuzzle).

| Model | Response Type | MathVista | MathVision | CVBench | Puzzle | AlgoPuzzle |
|---|---|---|---|---|---|---|
| | w/ Thinking | 44.90 | **16.45** | 70.36 | 52.50 | 24.78 |
| 2B | w/o Thinking | 41.9 | 15.79 | 71.72 | 62.90 | 27.50 |
| | No-Thinking-RFT | **48.80** | 13.16 | **76.76** | **70.85** | **29.17** |
| | w/ Thinking | **64.60** | 21.71 | 80.36 | 66.60 | 24.78 |
| 7B | w/o Thinking | 61.20 | **22.04** | **81.62** | **76.85** | **27.95** |
| | No-Thinking-RFT | 59.10 | 18.09 | 80.67 | 80.65 | **29.39** |

Table 18: Performance comparison of DPO, Listwise DPO, Thinking-RFT, and No-Thinking-RFT.

| Model | FGVC | MathVista | MathVision |
|---|---|---|---|
| DPO random | 46.23 | 40.15 | 12.3 |
| DPO model | 51.25 | 40.32 | 14.4 |
| Listwise DPO random | 47.19 | 40.78 | 12.45 |
| Listwise DPO model | 51.36 | 42.66 | 15.1 |
| Thinking-RFT | 74.41 | 44.90 | 16.45 |
| No-Thinking-RFT | 74.41 | 48.80 | 13.49 |

No-Thinking-RFT. This finding highlights the importance of online sampling and policy-gradient optimization in RFT. Moreover, model-generated negative samples consistently outperform random negatives in both DPO and listwise DPO settings, demonstrating that the quality of negative samples is critical to DPO training.

# I  Examples of the Model Response

In this subsection, we delve into the content of the thought process in Thinking-RFT among different model sizes and tasks. Specifically, we show We compare the response examples of four different thinking strategies in Figure 21~Figure 25, the examples of trivial reasoning responses of 2B models in Figure 15 ~ Figure 20, the inconsistent responses of 7B models in Figure 9~Figure 14. Typically, for 2B model responses, the content found in the "thinking" tags are somewhat trivial, such as "This is a photo of <class>" (classification) or "To find the area of the overlap between the two squares, we need to find the length of the diagonal of the smaller square." (MathVision) which offers little to no benefit towards arriving at the final answer, or they might already represent the final answers themselves without any reasoning process. For 7B models, the content within the thinking tag is significantly more meaningful and contributes more effectively to the reasoning process leading to the final answers. However, we frequently observe inconsistencies in the responses, where the content within the thinking tag diverges significantly from that in the answer tag. For example, the reasoning may support choice B, while the final answer provided is choice A. To quantify this issue, we use GPT-4o to extract the answers from thinking tag and answer tag and then calculate the proportion of such inconsistencies across each dataset. The detailed results are presented in Figure 2. Our analysis reveals that reasoning tasks are particularly prone to inconsistencies between the thinking and answer tags. Notably, as illustrated in Figure 2, the average accuracy of the answer tags in inconsistent responses is substantially higher than that of their corresponding thinking tags. This observation is expected, given that the answer tags are directly optimized via an accuracy-based reward signal, whereas the thinking tags receive no explicit supervision. Additionally, we find that the average accuracy of answer tags in inconsistent responses is lower than the overall average accuracy across all responses. This indicates that maintaining internal consistency between reasoning and final answers is conducive to improved model performance.

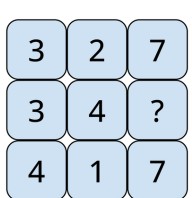

**Question:** What is the missing number of the part denoted with a question mark? Please choose your answer from: ['1', '2', '7', '5']

**Ground Truth: 5.**

**Response from Thinking-RFT:**

<think> The pattern in the rows and columns is as follows:
- First row: 3, 2, 7
- Second row: 3, 4, ?
- Third row: 4, 1, 7
The pattern in the rows and columns is that each number in the second row is the sum of the corresponding numbers in the first and third rows.
So, for the second row:
- 3 + 4 = 7
- 4 + 1 = 5
Therefore, the missing number in the second row is 5.
So, the answer is 5. </think>
<answer>1</answer>

---

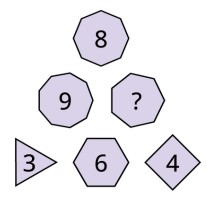

**Question:** What is the missing number of the part denoted with a question mark? Please choose your answer from: [7, 4, 5, 3]

**Ground Truth: 7.**

**Response from Thinking-RFT:**

<think> The pattern in the shapes is that the number of sides of each shape increases by one from the bottom to the top. The shapes are a triangle (3 sides), a hexagon (6 sides), and an octagon (8 sides). Following this pattern, the next shape should have 9 sides. The number in the center of the octagon is 8, so the missing number in the center of the pentagon should be 7. Therefore, the missing number is 7. </think>
<answer>5</answer>

**Response from No-Thinking-RFT:** 7.

Figure 9: Examples of the inconsistency error of fine-tuned Qwen-VL-2-7B model on PuzzleVQA. The correct answer part is marked as green, and wrong answer part is marked as red.

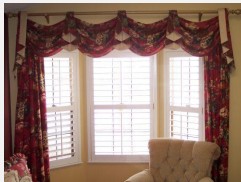

**Question:** How many walls are in the image? A. 1 B. 3 C. 2 D. 0
**Ground Truth: A.**
**Response from Thinking-RFT:**
<think> The image shows a window with curtains on both sides. The curtains are hanging from a rod that is attached to the wall. There is no visible wall in the image, only the window and the curtains. Therefore, the number of walls in the image is zero.</think>
<answer> (A) 1 </answer>

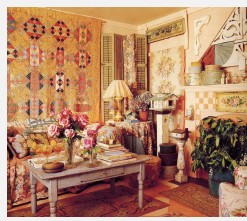

**Question:** How many rugs are in the image? A.3 B. 1 C. 0 D. 2
**Ground Truth: B.**
**Response from Thinking-RFT:**
<think> The image shows a living room with various items and decorations. There is a rug on the floor in front of the coffee table, and another rug on the floor in front of the fireplace. Therefore, there are two rugs in the image.</think> <answer> (B) 1 </answer>

Figure 10: Examples of the inconsistency error of fine-tuned Qwen-VL-2-7B model on CVBench. The correct answer part is marked as green, and wrong answer part is marked as red.

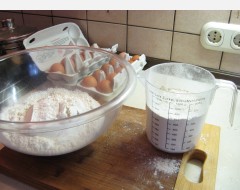

**Question:** Hint: Please answer the question requiring an integer answer and provide the final value, e.g., 1, 2, 3, at the end.
Question: what is the total volume of the measuring cup? (Unit: g)
**Ground Truth: 1000.**
**Response from Thinking-RFT:**
<think> The measuring cup has a scale marked in milliliters (mL). To convert milliliters to grams, we need to know the density of the substance being measured. For flour, the density is approximately 0.5 g/mL. The total volume of the measuring cup is 1000 mL. Therefore, the total volume in grams is 1000 mL * 0.5 g/mL = 500 g. </think>
<answer> 1000 </answer>

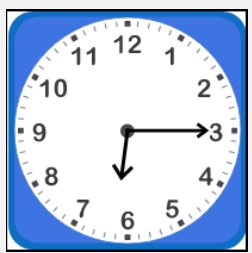

**Question:** Hint: Please answer the question and provide the correct option letter, e.g., A, B, C, D, at the end.
Question: What time is shown? Answer by typing a time word, not a number. It is (_) past six.
Choices:
(A) half
(B) quarter
(C) o'clock
(D) quarter to
(E) quarter past
**Ground Truth: B.**
**Response from Thinking-RFT:**
<think> The clock shows a time where the minute hand is pointing at the 3, which represents 15 minutes past the hour. The hour hand is between the 6 and the 7, which means it is 6 o'clock. Therefore, the time is 6:15. </think> <answer> A </answer>

Figure 11: Examples of the inconsistency error of fine-tuned Qwen-VL-2-7B model on MathVista. The correct answer part is marked as green, and wrong answer part is marked as red.

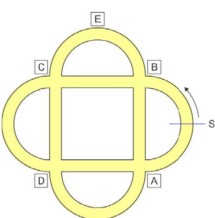

**Question:** Hint: Please answer the question and provide the correct option letter, e.g., A, B, C, D, E, F at the end.

Peter rides his bike along a cycle path in a park. He starts at point $S$ and rides in the direction of the arrow. At the first crossing he turns right, then at the next left, and then again to the right and then again to left. Which crossing does he not reach?
<image1>Choices: A. A B. B C. C D. D E. E

**Ground Truth: D.**

**Response from Thinking-RFT:**

<think> The arrow indicates the direction Peter rides his bike. He starts at point S and follows the path indicated by the arrow. He turns right at the first crossing, then left at the next, right again, and finally left. This means he will pass through the crossings labeled B, C, and D. The crossing labeled A is not mentioned in his path. Therefore, the crossing he does not reach is A.</think> <answer> C </answer>

---

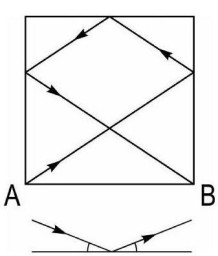

**Question:** Hint: Please answer the question and provide the correct option letter, e.g., A, B, C, D, E, F at the end.

A (very small) ball is kicked off from point A on a square billiard table with side length $2$ m. After moving along the shown path and touching the sides three times as indicated, the path ends in point $B$. How long is the path that the bal travels from A to B? (As indicated on the right: incident angle = emergent angle.)

Choices: A. 7 B. $2\sqrt{13}$ C. 8 D. $4\sqrt{3}$ E. $2 \cdot (\sqrt{2} + \sqrt{3})$

**Ground Truth: B.**

**Response from Thinking-RFT:**

<think> The path of the ball is a series of right angles, so we can break it down into segments of equal length. Each segment is the hypotenuse of a right triangle with legs of length 1 meter (since the side length of the square is 2 meters). Using the Pythagorean theorem, the length of each segment is $\sqrt{1^2 + 1^2} = \sqrt{2}$ meters. Since there are four segments, the total length of the path is $4\sqrt{2}$ meters. However, we can simplify this to $2\sqrt{2} \cdot 2 = 4\sqrt{2}$ meters. Therefore, the correct answer is D. </think>
<answer> B </answer>

Figure 12: Examples of the inconsistency error of fine-tuned Qwen-VL-2-7B model on MathVision. The correct answer part is marked as green, and wrong answer part is marked as red.

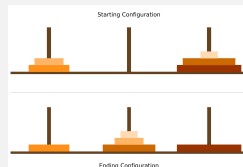

**Question:** You are playing a Tower of Hanoi game with 3 rods and 5 disks of various diameters, which can slide onto any rod. You are given the starting and ending configuration of the game as shown in the top and the bottom of the image, respectively. The game has the following rules: i) Only one disk may be moved at a time; ii) Each move consists of taking the upper disk from one of the stacks and placing it on top of another stack or on an empty rod; and iii) No disk can be placed on top of a disk that is smaller than it. What is the minimum number of moves required to go from the starting to the ending configuration? Please choose your answer from: 6, 2, 4, 5.

**Ground Truth: 5.**

**Response from Thinking-RFT:**

<think> The Tower of Hanoi problem is a classic puzzle that requires moving a stack of disks from one rod to another, following specific rules. The minimum number of moves required to solve the puzzle can be calculated using the formula $2^n - 1$, where n is the number of disks.

In this case, we have 5 disks, so the minimum number of moves required is $2^5 - 1 = 32 - 1 = 31$.

However, the options provided are 6, 2, 4, and 5. It seems there might be a misunderstanding in the question or the options. The correct answer based on the formula is 31, but it is not listed among the options.

Let's re-examine the image to see if there's any additional information or a different interpretation of the problem that might lead to a different answer.

Upon closer inspection, the image shows a specific starting and ending configuration, but it does not provide enough information to determine the exact number of moves required to solve the puzzle. The image does not show the intermediate steps or the specific rules applied to reach the ending configuration.

Given the options provided, it seems there might be a misunderstanding or an error in the question. The correct answer based on the formula is 31, but it is not listed among the options.

Therefore, the minimum number of moves required to go from the starting to the ending configuration is 31, but this is not listed among the options.

Final answer: 31 (not listed among the options)</think>

<answer>5</answer>

---

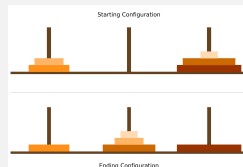

**Question:** The image shows the calendar of a month of a particular non-leap year. Which day of the week was on February 25 of that year?
Please choose your answer from: Friday, Sunday, Monday, Tuesday.

**Ground Truth: Friday.**

**Response from Thinking-RFT:**

<think> The calendar shows that February 25 is a Friday. Therefore, the day of the week for February 25 of that year is Friday. </think>

<answer> Tuesday </answer>

Figure 13: Examples of the inconsistency error of fine-tuned Qwen-VL-2-7B model on AlgoPuzzleVQA. The correct answer part is marked as green, and wrong answer part is marked as red.

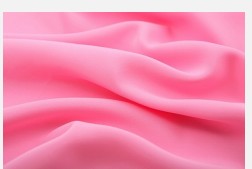

**Question:** What type of texture is in the photo?
Please choose one from list [ matted, grid, crosshatched, frilly, flecked, zigzagged, freckled, waffled, chequered, porous, striped, fibrous, sprinkled, perforated, crystalline, cracked, banded, lacelike, veined, polka-dotted, paisley, scaly, pleated, smeared, dotted, cobwebbed, wrinkled, gauzy, interlaced, knitted].
**Ground Truth: wrinkled.**
**Response from Thinking-RFT:**
<think> The photo shows a close-up of a fabric with a smooth, continuous surface. There are no visible patterns, lines, or textures that would suggest a grid, crosshatched, frilly, flecked, zigzagged, freckled, waffled, chequered, porous, striped, fibrous, sprinkled, perforated, crystalline, cracked, banded, lacelike, veined, polka-dotted, paisley, scaly, pleated, smeared, dotted, cobwebbed, wrinkled, gauzy, interlaced, or knitted texture. The fabric appears to be soft and smooth, with a consistent, even texture. Therefore, the most appropriate texture to describe this fabric is "pleated".</think> <answer> wrinkled </answer>

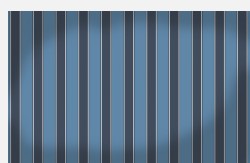

**Question:** What type of texture is in the photo?
Please choose one from list [ smeared, bubbly, fibrous, lacelike, frilly, potholed, spiralled, chequered, flecked, paisley, stained, meshed, wrinkled, zigzagged, polka-dotted, matted, freckled, lined, stratified, banded, cracked, sprinkled, gauzy, striped, marbled, crosshatched, swirly, pitted, grid, veined].
**Ground Truth: striped.**
**Response from Thinking-RFT:**
<think> <think> The photo shows a pattern of vertical lines. The lines are straight and evenly spaced, creating a consistent and repetitive visual effect. There are no visible curves, ripples, or other irregularities in the lines. Therefore, the texture in the photo is best described as "lined." </think> <answer> striped </answer>

Figure 14: Examples of the inconsistency error of fine-tuned Qwen-VL-2-7B model on classification DTD dataset. The correct answer part is marked as green, and wrong answer part is marked as red.

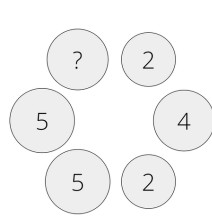

**Question:** How many cubes have been taken from the block?
**Ground Truth: 4.**
**Response from Thinking-RFT:**
<think>
To solve this, we need to determine the missing number in the pattern. Let's examine the numbers and their positions.
</think>
<answer>4</answer>

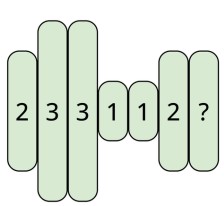

**Question:** What is the missing number of the part denoted with a question mark?
Please choose your answer from: [2, 3, 4, 1]
**Ground Truth: 2.**
**Response from Thinking-RFT:**
<think>
To solve this, we need to determine the missing number in the sequence. Let's analyze the pattern in the sequence: 2, 3, 3, 1, 1, 1, 2.
</think>

<answer>
1
</answer>

Figure 15: Examples of the trivial reasoning of fine-tuned Qwen-VL-2-2B model on PuzzleVQA.

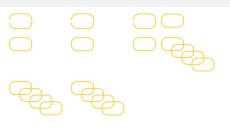

**Question:** Alice has 10 segments of chains of different lengths as shown in the image. The total length of all the segments combined is 19 pieces. She has a saw machine with which a closed piece can be cut opened. She also has a welding machine with which an open piece can be closed. Each cut takes 5 minutes and each welding takes 5 minutes. Initially, she has 3 segments each with 1 open piece as shown in the image. All the other pieces are closed. She now wants to make the longest possible necklace using all the available 19 pieces. Each piece in the necklace would be connected to exactly two other pieces. This would require cutting open some pieces and then joining all the resulting segments together. What is the minimum time in which she can create the necklace?
Please choose your answer from: 40, 41, 35, 3.
**Ground Truth: 35.**
**Response from Thinking-RFT:**
<think>
To create the longest possible necklace, Alice needs to maximize the number of segments she can use. She has 10 segments, and she wants to use all 19 pieces. The goal is to maximize the number of segments used while ensuring that each segment is connected to exactly two other segments.
</think>

<answer>35</answer>

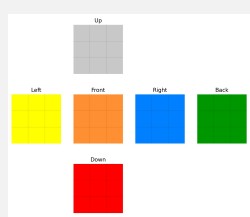

**Question:** A 3 * 3 Rubik's Cube has six different coloured panels: red, green, blue, yellow, orange, and grey. The initial state of the cube in terms of the different colour positions in its six faces is shown in the image. To represent the movements of the cube we use six letters: U for Up, D for Down, L for Left, R for Right, F for Front, B for Back. These letters are used in sequence where you need to perform each letter in the sequence from left to right. Each letter tells you to move that face clockwise by 90 degrees. A number 'n' immediately after a letter denotes that you need to move that face clockwise by 90 * n degrees. For example, 'U R3' would mean rotating the up face 90 degrees clockwise and then rotating the right face 270 degrees clockwise. You perform the move sequence 'B B2' starting from the state shown in the image. What would be the number of small 1 * 1 red squares in the left face after completing the move sequence?
Please choose your answer from: 3, 4, 7, 0.
**Ground Truth: 3.**
**Response from Thinking-RFT:**
<think>
To solve this problem, we need to understand the sequence of moves and the effect each move has on the cube's state. The sequence is 'B B2', which means rotating the left face clockwise by 90 degrees twice. We need to calculate the final state of the cube after these moves.
</think>

<answer>7</answer>

Figure 16: Examples of the trivial reasoning of fine-tuned Qwen-VL-2-2B model on AlgoPuzzleVQA.

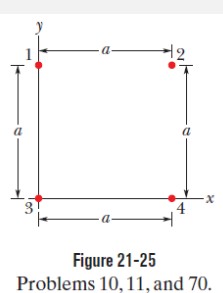

**Figure 21-25**
Problems 10, 11, and 70.

**Question:** Hint: Please answer the question requiring a floating-point number with two decimal places and provide the final value, e.g., 1.23, 1.34, 1.45, at the end.
Question: In Fig. 21-25, the particles have charges $q_1 = -q_2 = 100$nC and $q_3 = -q_4 = 200$nC, and distance $a = 5.0$ cm. What is the $x$ component of the net electrostatic force on particle 3?
**Ground Truth: 0.17.**
**Response from Thinking-RFT:**
<think>
To find the x component of the net electrostatic force on particle 3, we need to find the net force acting on particle 3 and then divide it by the charge of particle 3.
</think>
<answer>100</answer>

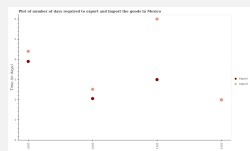

**Question:** Hint: Please answer the question requiring an integer answer and provide the final value, e.g., 1, 2, 3, at the end.
Question: What is the difference between the highest and the lowest time required to import ?
**Ground Truth: 4.**
**Response from Thinking-RFT:**
<think>
To determine the difference between the highest and the lowest time required to import goods in Mexico, we need to analyze the data provided in the plot.
</think>

<answer>4</answer>

Figure 17: Examples of the trivial reasoning of fine-tuned Qwen-VL-2-2B model on MathVista.

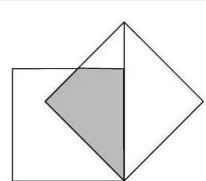

**Question:** Hint: Please answer the question and provide the correct option letter, e.g., A, B, C, D, E, F at the end.

Two squares with side 1 have a common vertex, and the edge of one of them lies along the diagonal of the other. What is the area of the overlap between the squares? <image1>Choices: A. $\sqrt{2} - 1$

B. $\frac{\sqrt{2}}{2}$

C. $\frac{\sqrt{2}+1}{2}$

D. $\sqrt{2} + 1$

E. $\sqrt{3} - \sqrt{2}$

**Ground Truth: A.**
**Response from Thinking-RFT:**
<think>
To find the area of the overlap between the two squares, we need to find the length of the diagonal of the smaller square.
</think>

<answer>B</answer>

---

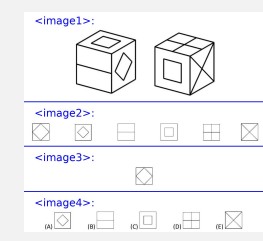

**Question:** <image1>
In the picture above we see a cube in two different positions.
The six sides of the cube look like this:
<image2>
Which side is opposite to <image3>?
<image4>
**Ground Truth: C.**
**Response from Thinking-RFT:**
<think>
To determine which side is opposite to <image3>, we need to identify the opposite side of the cube in <image4>.
</think>

<answer>D</answer>

Figure 18: Examples of the trivial reasoning of fine-tuned Qwen-VL-2-2B model on MathVision.

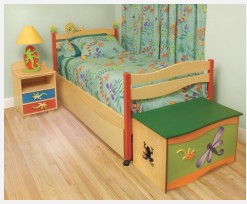

**Question:** How many chests are in the image? A. 3 B. 1 C. 2 D. 0
**Ground Truth: B.**
**Response from Thinking-RFT:**
<think>
The image shows a bed with a nightstand, a curtain, and a toy chest. There is only one chest in the image.
</think>
<answer> 1 </answer>

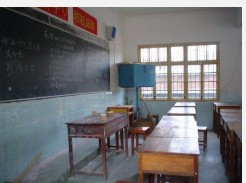

**Question:** How many walls are in the image? A. 1 B. 0 C. 3 D. 4 E. 2
**Ground Truth: E.**
**Response from Thinking-RFT:**
<think>
The image shows a classroom with a blackboard, desks, and windows. The walls are painted in a light color, and there are no visible windows on the back wall. Therefore, the correct answer is (A) 1.
</think>
<answer> 1 </answer>

Figure 19: Examples of the reasoning of fine-tuned Qwen-VL-2-2B model on CVBench. Unlike the trivial responses in reasoning tasks, the thinking process in CVBench is not trivial.

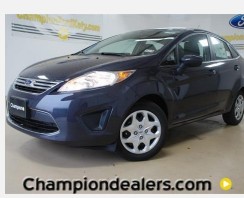

**Question:** What type of object is in the photo? Please choose one from list [ 2012 Acura TL Sedan, 2012 Chrysler Town and Country Minivan,...]
**Response from Thinking-RFT:**
<think> The object in the photo is a 2012 Ford Fiesta Sedan.</think>
<answer> 2012 Ford Fiesta Sedan</answer>



**Question:** What type of texture is in the photo? Please choose one from list [ spiralled, knitted, sprinkled....]
**Response from Thinking-RFT:**
<think> The texture in the photo appears to be woven. </think>
<answer>Woven</answer>

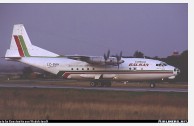

**Question:** What type of aircraft is in the photo? Please choose one from list [ MD-90, Beechcraft 1900, DHC-6, MD-87, 777-300, CRJ-900...]
**Response from Thinking-RFT:** <think> The aircraft in the photo is an An-12. </think>
<answer>An-12</answer>

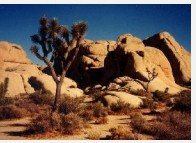

**Question:** What type of object is in the photo? Please choose one from list [ okapi, bass, revolver, cup, nautilus, ...]
**Response from Thinking-RFT:**
<think> The object in the photo is a Joshua tree. </think>
<answer>Joshua_tree</answer>

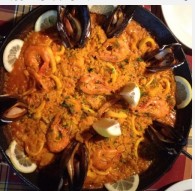

**Question:** What type of object is in the photo? Please choose one from list [ grilled_cheese_sandwich, caesar_salad, lasagna, ...]
**Response from Thinking-RFT:**
<think> The photo shows a dish of paella, which is a traditional Spanish dish made with rice, seafood, vegetables, and spices.</think>
<answer>Paella</answer>

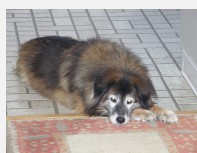

**Question:** What type of object is in the photo? Please choose one from list [ miniature_pinscher, yorkshire_terrier,...]
**Response from Thinking-RFT:**
<think>The dog in the photo appears to be a Leonberger, a breed known for its long, thick coat and large size. </think>
<answer>Leonberger</answer>

Figure 20: Examples of Thinking-RFT fine-tuned model responses across six datasets. The thinking content appears not very beneficial to deriving the final answers.

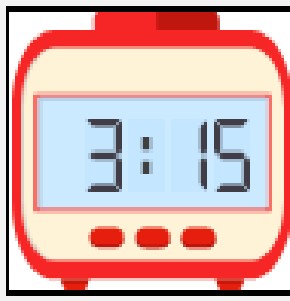

**Question:** Hint: Please answer the question and provide the correct option letter, e.g., A, B, C, D, at the end.

Question: What time is shown? Answer by typing a time word, not a number. It is (_) past three.

Choices:

(A) half

(B) quarter

(C) o'clock

(D) quarter to

(E) quarter past

**Ground Truth: B.**

**Response from 2B Thinking-RFT:** <think>

To determine the correct time shown on the clock, let's analyze the image:

- The clock is red and has the time displayed as 3:15.

- The time is shown as "quarter to" because it is 15 minutes past the hour.

</think>

<answer> D </answer>

**Response from 2B Think-After-Answer:** <answer> E </answer>

<reason> The time shown is quarter past, as indicated by the "15" in the clock. The word "quarter" is used to describe the position of the hour hand between the two halves of the clock. Therefore, the correct option is E. </reason>

**Response from 2B No-Thinking-RFT:** D

**Response from 2B Adaptive-Thinking:** <answer> E </answer>

**Response from 7B Thinking-RFT:** <think> The time shown on the clock is 3:15. To determine the time, we look at the minute hand, which is pointing at 3, and the hour hand, which is between 3 and 4. Since the minute hand is at 15 minutes past the hour, it is 15 minutes past 3. Therefore, the correct answer is "quarter past" because 15 minutes is a quarter of an hour. </think> <answer> B </answer>

**Response from 7B Think-After-Answer:** <answer> B </answer>

<reason> The time shown on the clock is 3:15. Since the minute hand is on the 3, which represents 15 minutes past the hour, the correct answer is "quarter past" or "quarter to". However, since the question asks for a time word, "quarter past" is the appropriate choice. </reason>

**Response from 7B No-Thinking-RFT:** B

**Response from 7B Adaptive-Thinking:** <think> The time shown is 3:15. The minute part is 15, which is a quarter of an hour. Therefore, the time is 15 minutes past 3. </think> <answer> B </answer>

Figure 21: Comparison of the model response of different fine-tuned model on MathVista.

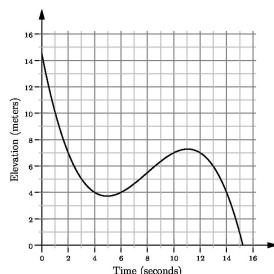

**Question:** Malaika is skiing on a mountain. The graph below shows her elevation, in meters, above the base of the mountain as she skis along a trail. In total, how many seconds does she spend at an elevation between 4 and 7 meters?

**Ground Truth: 8.**

**Response from 2B Thinking-RFT:** <think>
To find out how many seconds Malaika spends at an elevation between 4 and 7 meters, we need to find the area under the curve between these two values.
</think>
<answer>10</answer>

**Response from 2B Think-After-Answer:** <answer>10</answer>
<reason>She spends 10 seconds between 4 and 7 meters. The graph shows that she is at an elevation between 4 and 7 meters for 10 seconds.</reason>

**Response from 2B No-Thinking-RFT:** 10

**Response from 2B Adaptive-Thinking:** <answer> 10 </answer>

**Response from 7B Thinking-RFT:** <think> The graph shows the elevation of Malaika over time. To find out how many seconds she spends at an elevation between 4 and 7 meters, we need to look at the portion of the graph where the elevation is between 4 and 7 meters. This is the area under the curve between the horizontal lines at 4 meters and 7 meters. By examining the graph, we can see that Malaika spends approximately 6 seconds at an elevation between 4 and 7 meters. </think> <answer> 6 </answer>

**Response from 7B Think-After-Answer:** <answer>6</answer>
<reason>The graph shows that Malaika's elevation is between 4 and 7 meters for 6 seconds. This is the time when the line is above the 4-meter mark and below the 7-meter mark on the graph. </reason>

**Response from 7B No-Thinking-RFT:** 6

**Response from 7B Adaptive-Thinking:** <think> The graph shows Malaika's elevation over time. To find the time she spends at an elevation between 4 and 7 meters, we need to look at the portion of the graph where the elevation is between 4 and 7 meters. This portion of the graph is between the time of 4 seconds and 12 seconds. Therefore, Malaika spends 8 seconds at an elevation between 4 and 7 meters. </think> <answer> 8 </answer>

Figure 22: Comparison of the model response of different fine-tuned model on MathVision.

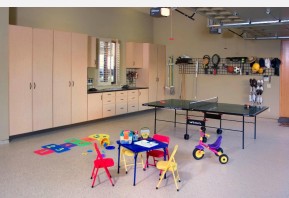

**Question:** How many light switchs are in the image?
A. 1 B. 0 C. 2 D. 3
**Ground Truth: A.**
**Response from 2B Thinking-RFT:**
<think>
To determine the number of light switchs in the image, let's analyze the visible parts of the room:

1. The wall on the left side has several light switchs.
2. The wall on the right side has several light switchs.
3. The ceiling has several light switchs.
4. The floor has no visible light switchs.

Based on this analysis, there are 4 light switchs in the image.
</think>

<answer> (C) 2 </answer>
**Response from 2B Think-After-Answer:**
<answer>1</answer>
<reason>There is only one light switch visible in the image, which is located on the wall to the right of the garage area.</reason>
**Response from 2B No-Thinking-RFT:** (A) 1
**Response from 2B Adaptive-Thinking:** <answer>1</answer>
**Response from 7B Thinking-RFT:**
<think>There are two light switches visible in the image, one on the wall near the ceiling and another on the wall near the floor. Therefore, the total number of light switches is 2.</think>
<answer>(A) 1</answer>
**Response from 7B Think-After-Answer:**
<answer> (A) 1 </answer>
<reason> There is one light switch visible on the wall in the image. </reason>
**Response from 7B No-Thinking-RFT:** (A) 1
**Response from 7B Adaptive-Thinking:**
<think> There are two light switches in the image, one on the wall near the ceiling and another on the wall near the floor. </think> <answer> A </answer>

Figure 23: Comparison of the model response of different fine-tuned model on CVBench.

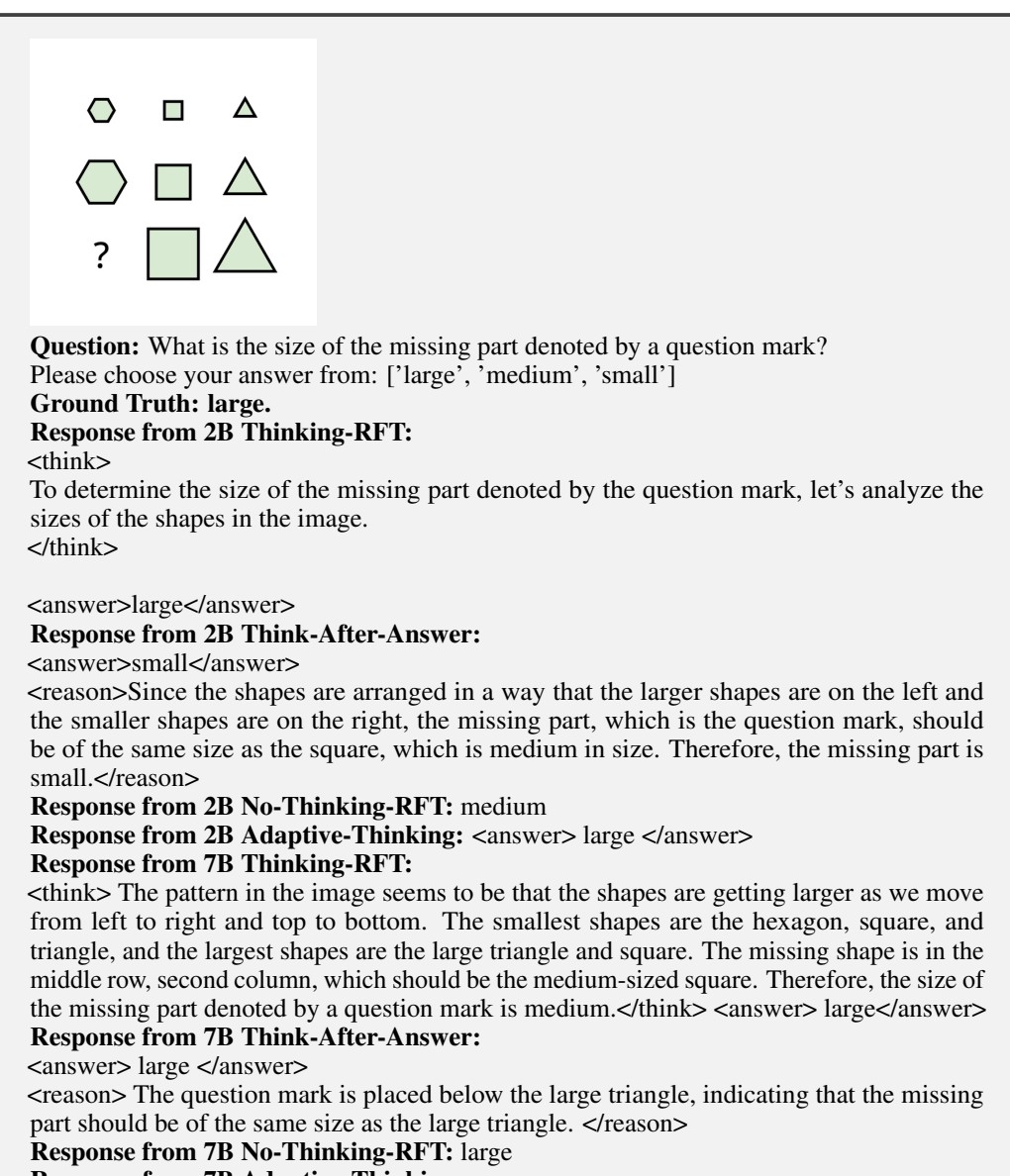

**Question:** What is the size of the missing part denoted by a question mark?
Please choose your answer from: ['large', 'medium', 'small']
**Ground Truth: large.**
**Response from 2B Thinking-RFT:**
<think>
To determine the size of the missing part denoted by the question mark, let's analyze the sizes of the shapes in the image.
</think>

<answer>large</answer>
**Response from 2B Think-After-Answer:**
<answer>small</answer>
<reason>Since the shapes are arranged in a way that the larger shapes are on the left and the smaller shapes are on the right, the missing part, which is the question mark, should be of the same size as the square, which is medium in size. Therefore, the missing part is small.</reason>
**Response from 2B No-Thinking-RFT:** medium
**Response from 2B Adaptive-Thinking:** <answer> large </answer>
**Response from 7B Thinking-RFT:**
<think> The pattern in the image seems to be that the shapes are getting larger as we move from left to right and top to bottom. The smallest shapes are the hexagon, square, and triangle, and the largest shapes are the large triangle and square. The missing shape is in the middle row, second column, which should be the medium-sized square. Therefore, the size of the missing part denoted by a question mark is medium.</think> <answer> large</answer>
**Response from 7B Think-After-Answer:**
<answer> large </answer>
<reason> The question mark is placed below the large triangle, indicating that the missing part should be of the same size as the large triangle. </reason>
**Response from 7B No-Thinking-RFT:** large
**Response from 7B Adaptive-Thinking:**
<answer> large </answer>

Figure 24: Comparison of the model response of different fine-tuned model on PuzzleVQA.

**Question:** The board shown in the image is a sliding puzzle of 5 * 5 tile dimensions. It has 24 numbered tiles and one unoccupied (open) position. Tiles in the same row or column of the open position can be moved by sliding them horizontally or vertically, respectively. All tiles always stay and move inside the red boundary wall, as shown in the image. A move is defined as moving the open position by one tile unit in any available direction. You start from the board position shown in the image and perform exactly 2 moves. How many unique final board positions can you reach?

Please choose your answer from: 3, 1, 4, 5.

**Ground Truth: 5.**

**Response from 2B Thinking-RFT:** <think>To determine the number of unique final board positions that can be reached by performing exactly 2 moves, we need to analyze the possible moves that can be made from the given starting position and the constraints imposed by the sliding puzzle rules. </think><answer>5</answer>

**Response from 2B Think-After-Answer:** <answer>5</answer>
<reason>After performing 2 moves, the board can reach 5 unique final positions: (1, 1), (1, 2), (1, 3), (1, 4), and (1, 5). These positions are reached by sliding the open position in the first row and column of the board. The other positions are not reachable because the board cannot move the open position in the first row or column of the board. Therefore, there are 5 unique final board positions.</reason>

**Response from 2B No-Thinking-RFT:** 5

**Response from 2B Adaptive-Thinking:** <answer> 5 </answer>

**Response from 7B Thinking-RFT:** <think> The puzzle is a 5x5 sliding puzzle with 24 numbered tiles and one empty space. We need to find the number of unique final board positions after exactly 2 moves. Let's analyze the possible moves:
1. Moving the open position to the left or right by one tile unit.
2. Moving the open position up or down by one tile unit.
Since the puzzle is 5x5, there are 4 possible moves for each direction (left, right, up, down). Therefore, there are a total of 8 possible moves. However, we need to perform exactly 2 moves. The number of ways to choose 2 moves out of 8 is given by the combination formula C(n, k) = n! / (k!(n-k)!), where n is the total number of moves and k is the number of moves we want to choose. So, C(8, 2) = 8! / (2!(8-2)!) = 28. However, we need to consider that some of these combinations might result in the same final board position. For example, moving the open position to the left and then to the right is the same as moving it to the right and then to the left. Therefore, we need to remove these duplicates.
To do this, we can use the concept of permutations. The number of permutations of 8 items taken 2 at a time is given by P(n, k) = n! / (n-k)!, where n is the total number of items and k is the number of items to choose. So, P(8, 2) = 8! / (8-2)! = 56. However, we need to divide this by 2 to account for the duplicates mentioned earlier. Therefore, the number of unique final board positions is 28 / 2 = 14. So, the answer is 14.</think> <answer>5</answer>

**Response from 7B Think-After-Answer:** <answer> 4 </answer>
<reason> The open position can move to any of the four adjacent tiles (up, down, left, right) in one move. Since there are 24 tiles and each tile can be in one of the 25 positions (including the open position), there are 24 possible starting positions for the open position. After the first move, there are 4 possible positions for the open position (since it can move to any of the four adjacent tiles). Therefore, there are 24 * 4 = 96 possible final board positions. However, since the open position can move in any of the four directions, there are 4 * 4 = 16 possible final board positions. </reason>

**Response from 7B No-Thinking-RFT:** 4

**Response from 7B Adaptive-Thinking:** <answer> 4 </answer>

Figure 25: Comparison of the model response of different fine-tuned model on AlgoPuzzleVQA.

