# OpenReview forum: "To Think or Not To Think: A Study of Thinking in Rule-Based Visual Reinforcement Fine-Tuning"
_NeurIPS.cc/2025/Conference — NeurIPS 2025 spotlight_

### Official Review · Reviewer_rKqb · 2025-07-01

**Clarity:** 4
**Significance:** 4
**Originality:** 3
**Rating:** 5
**Confidence:** 4

**Summary:**

The author investigates whether the "think" process is essential during the Rejection-Fine-Tuning (RFT) stage. Models that undergo Reinforcement Learning from Verifier Feedback (RLVR) typically show significant improvements in reasoning performance, a gain often attributed to the establishment of a stable thinking process. However, the author finds that even when the model is not required to output a "think" process during training, it can achieve performance comparable to conventional RLVR. The topic explored in this paper is very interesting, and the experiments are comprehensive.

**Questions:**

Please refer to the weakness

**Ethical Concerns:**

["NO or VERY MINOR ethics concerns only"]

**Final Justification:**

My major concerns have been well addressed, thus I would like to raise my score.

**Quality:**

3

**Strengths And Weaknesses:**

### Strength
- I quite like the motivation of this paper.
- The paper explores many aspects and challenges the community's belief that the "think" process is the primary source of gains from RFT. The experiments are solid.


### Weaknesses

1.  The use of a 4-shot training setting (i.e., 4 examples per class), especially when sufficient data is available, seems somewhat "toy-like." This small-data regime is also not the typical scenario where Supervised Fine-Tuning (SFT) is claimed to be most effective.

2.  Although this work largely argues that the "think" process is not essential, it can also be seen as fundamentally challenging the RFT process itself. The current characteristics of RFT can be broken down into the following points:
    - The model samples both "think" and "answer".
    - A verifiable reward that is not easily "hacked."
    - Learning from a set of positive and negative samples.

    If the "think" process is removed from RFT, the model only needs to sample the "answer." For training samples with verifiable rewards (common in domains like math or code), constructing answers is straightforward. For example, given the problem "1+1 = ?", we can easily generate answers from the range [1-10], treating "2" as the positive sample and the rest as negative samples. By applying a list-wise DPO algorithm, one could achieve a similar effect to "no-think RFT." Since offline DPO does not involve online sampling, its cost is much lower than that of "no-think RFT." This could potentially lead to a scenario where **offline DPO ≥ no-think RFT ≥ RFT**. I believe this comparative analysis would be very valuable, as this transitive relationship might suggest a new RL paradigm.

3.  There is another difference between "no-think RFT" and RFT that needs consideration. "no-think RFT" can essentially be considered a learning process with one positive example versus multiple negative examples. In contrast, RFT can generate diverse positive examples by combining various "think" processes with the correct answer. In "no-think RFT," however, the positive example is unique since it only consists of the final answer. This analytical experiment is also missing from the paper.

4.  Section 4.1 (L229-230) directly assumes that the "explicit" think process is harmful. However, based on the experimental data, this conclusion only seems to hold true for the small model (2B). In the experiments with the 7B model, although "no-think" scores higher than "think," the "adaptive think" mode, which clearly converges to a thinking pattern, achieves the highest score. This results in a relationship of `think < no-think < adaptive-think`. Or, given the small differences, it could be interpreted as `think ≈ no-think ≈ adaptive-think`. In either case, for the 7B model, one cannot conclude that the explicit "think" process is harmful.

5.  The significant differences between the 2B and 7B models—in their convergence trends for "adaptive think" and in the perceived harm of the explicit "think" process—are noteworthy. This raises the question of which of this paper's conclusions can be generalized to larger 32B or 72B models. If the trends for 2B and 7B were similar, we could be more optimistic that the paper's conclusions have good scaling properties.

6.  The "free lunch phenomenon" sounds strange. Isn't this just about performance transfer? I don't see this as a particularly novel finding. What is the author trying to convey in this section?

7.  Figure 6: A figure belonging in the main text is placed in the appendix?

---

> ### Author Rebuttal · Authors · 2025-07-31
>
> Thank you very much for the time and effort in reviewing our paper. We are encouraged that you found the motivation compelling and the experiments solid. We truly appreciate your thoughtful discussion of DPO and RFT.
>  Below are our responses (R) to the weaknesses (W) mentioned in your review.
>
> ------
>
> **W1: The use of a 4-shot training setting (i.e., 4 examples per class) seems somewhat "toy-like."**
>
> **R: (i)** We would like to respectfully clarify that few-shot image classification is a important multi-modal task for exploring the fine-tuning of vision-language models under limited labeled data, and has been widely studied in prior work [1, 2, 3]. This setting is intentionally designed to emphasize data efficiency and fine-tuning under minimal supervision. Therefore, it should not be regarded as a toy-like experimental setup.
>
> **(ii)** We would like to respectfully clarify that we did not claim SFT is most effective in the few-shot setting. As stated in lines 151–159, [4] observed that existing MLLMs struggle with classification, but performance can be improved through SFT with large-scale labeled datasets. However, such large-scale SFT is costly and time-consuming, particularly for RFT, which may require months of GPU hours on millions of labeled images. This exceeds our computational resources. Therefore, we focus on data-efficient strategies and explore few-shot fine-tuning instead.
>
> ------
>
> **W2: Discussion betwen offline DPO, No-Thinking-RFT, and Thinking-RFT.**
>
> **R:** This is an insightful point, and we sincerely thank the reviewer for raising it.
>
> **(i)** Although No-Thinking-RFT removes the “think” phase, a list-wise DPO algorithm **should not** be expected to reproduce similar effect due to the fundamental difference between DPO and GRPO. We discuss the detailed difference below:
>
> 1. **Sampling strategy.** List-wise DPO selects one positive and several negatives *offline*. In contrast, No-Thinking-RFT samples responses *online* at every training step; the mix of positives and negatives is not fixed at $1:(N−1)$ but varies with task difficulty and model competence (from $0:N$ to $N:0$ ). Thus while removing thinking, No-Thinking-RFT continues to explore and learn from fresh errors, whereas DPO does not.
> 2. **Gradient update rule.** No-Thinking-RFT computes a policy-gradient using verifiable rewards ($R=1$ for an exact match, $R=0$ otherwise) and a group-advantage formulation, i.e. GRPO. Listwise DPO, by contrast, minimises a cross-entropy loss over a pre-computed list, functioning more like contrastive SFT. This distinction yields different learning dynamics.
>
> Because of these differences, listwise DPO **should not** be expected to match the empirical behaviour of No-Thinking-RFT.
>
> **(ii)** We also implemented preliminary DPO and listwise DPO baselines to quantify the performance difference. We evaluated two negative-sampling strategies: one using samples generated by Qwen2-VL (DPO model) and the other using randomly generated samples (DPO random) as suggested by the reviewer.  Experiments were carried out on the FGVCAircraft classification dataset and the MathVQA dataset. The results shown in the table below indicate that both DPO variants under both sampling strategies perform substantially worse than Thinking-RFT and No-Thinking-RFT. This finding highlights the importance of online sampling and policy-gradient optimization in RFT. Moreover, model-generated negative samples consistently outperform random negatives in both DPO and listwise DPO settings, demonstrating that the quality of negative samples is critical to DPO training.
>
> We will add this discussion in the next version of paper.
>
> | Model               | FGVC  | MathVista | MathVision |
> | ------------------- | ----- | --------- | ---------- |
> | DPO random          | 46.23 | 40.15     | 12.3       |
> | DPO model           | 51.25 | 40.32     | 14.4       |
> | Listwise DPO random | 47.19 | 40.78     | 12.45      |
> | Listwise DPO model  | 51.36 | 42.66     | 15.1       |
> | Thinking-RFT        | 74.41 | 44.90     | 16.45      |
> | No-Thinking-RFT     | 74.41 | 48.80     | 13.49      |
>
> ------
>
> **W3: Positive-sample diversity differs between Thinking-RFT and No-Thinking-RFT.**
>
>  **R:** This is a valuable observation and we thank the reviewer for pointing it out. During fine-tuning, Thinking-RFT can indeed produce multiple reward-positive trajectories (and many more negatives) for the same verifiable answer, whereas No-Thinking-RFT yields only one positive sample—the exact answer. Such a disparity could lead the Thinking-RFT model to generate more diverse outputs at inference. To test this as an extra analytical experiment , we ran each fine-tuned model five times on MathVista with Qwen2-VL-2B and recorded mean accuracy, its standard deviation, and pass@5 accuracy in Table below. We can find that Thinking-RFT exhibits both a higher gain of pass@5 and a larger 5 run standard deviation than No-Thinking-RFT, confirming greater output diversity.
>
> | Model           | pass@1 | Mean $\pm$ std (5 runs) | pass@5 |
> | --------------- | ------ | ----------------------- | ------ |
> | Thinking-RFT    | 44.90  | 44.84$\pm$ 0.45         | 46.70  |
> | No-Thinking-RFT | 48.80  | 48.74$\pm$ 0.15         | 48.90  |
>
> ------
>
> **W4: Your claim that explicit thinking is harmful relies mainly on the 2B results; for 7B the ranking is think < no-think < adaptive-think (or roughly equal across settings), so the data do not justify calling explicit thinking harmful for larger models.**
>
>  **R:** We apologize for the confusion. Lines 229-230 contain a wording mistake: we did not intend to claim that explicit thinking is harmful for *all* model sizes. We want to use Think-After-Answer to show that explicit thinking is detrimental for the 2B model (Finding 1 that 2B No-Thinking-RFT surpasses Thinking-RFT) and not universally necessary, though not harmful, for the 7 B model (Finding 2 that 2B No-Thinking-RFT, Thinking-RFT, and Adaptive-Thinking have very similar results). We will revise the sentence to:
>
> > *“For the 2B model, the explicit ‘think’ process is harmful; for the 7B model it is not universally necessary.”*
>
> ------
>
> **W5: Uncertain scalability to $\ge$14B models.**
>
>  **R:** We acknowledge this limitation that the trend of Adaptive-Thinking differs for 2B and 7B models. Because of GPU constraints we cannot currently run higher-scale experiments such as Qwen2-VL-32B; our study therefore focuses on small (2B) and medium (7B) checkpoints, the sizes most practitioners can fine-tune. We already flag this as a limitation and explicitly leave larger-model validation as future work for teams with the necessary computational resources.
>
> ------
>
> **W6: Free-lunch phenomenon sounds trivial.**
>
>  **R:** We would like to respectfully clarify that observing performance *improvements* on *other distinct* datasets after fine-tuning on a single dataset is **not** trivial in few-shot image classification. Prior work [2] reports severe *catastrophic forgetting* problem in few-shot image classification learning: for example, CoCoOp fine-tuned on ImageNet reduces DTD accuracy from 76.4% to 45.7%. We observe a similar drop with Qwen2-VL SFT. By contrast, both Thinking-RFT and No-Thinking-RFT *avoid* forgetting and instead *increase* accuracy on the other datasets. This finding contrasts sharply with previous few-shot learning results and is therefore striking. We want to use this finding to convey that RFT not only avoids catastrophic forgetting but also teaches the model the underlying essence of classification rather than memorising a single dataset (Section 3.3). To avoid confusion, we will rename this finding the *'cross-dataset generalisation effect'*.
>
> ------
>
> **W7: Figure 6 belongs in the main text but in Appendix.**
>
>  **R:** Thank you for pointing that out. Due to page limitations, we placed Figure 6 in the Appendix. We will move it to the main text in the next version.
>
> ------
>
> **References**
>
> [1] Learning to Prompt for Vision-Language Models, IJCV 2022.
>
> [2] Conditional Prompt Learning for Vision-Language Models, CVPR 2022.
>
> [3] GraphAdapter: Tuning Vision-Language Models With Dual Knowledge Graph, NeurIPS 2023.
>
> [4] Why are visually grounded language models bad at image classification? NeurIPS 2024.

---

> > ### Comment · Reviewer_rKqb · 2025-08-05
> > **After reading the rebuttal**
> >
> > I would like to thank the authors for their detailed efforts during the rebuttal phase. Their response has successfully resolved my primary concerns. The subject of this paper is of considerable interest; to think or not to think is a significant question for the research community, particularly for industrial applications. Up to now, thinking is more preferred when solving complex math or code problems, but the necessity is still questionable.  Accordingly, I have raised my score to 5.

---

> > > ### Author Response · Authors · 2025-08-05
> > > **Thanks again for reviewing our paper!**
> > >
> > > We sincerely appreciate the time and effort you dedicated to reviewing our submission. Your insightful feedback is invaluable in helping us improve our work. Thank you once again for your support and thoughtful review.

---

### Official Review · Reviewer_wtZh · 2025-07-02

**Clarity:** 3
**Significance:** 2
**Originality:** 3
**Rating:** 4
**Confidence:** 3

**Summary:**

This paper first extends Thinking-RFT to visual tasks and then investigates whether explicit thinking is always necessary and beneficial in RFT. This is motivated by Deepseek R1 shows that explicit thinking can help model improve the performance. The authors propose No-Thinking-RFT on 2B and 7B models, demonstrating that explicit thinking does not consistently improve performance during RFT in visual perception tasks.

**Questions:**

1. The authors mention Deepseek-R1 as the motivation; however, I think Deepseek-R1 does not emphasize the benefits of explicit thinking specifically in small-scale models. Since the experiments presented in this paper focus exclusively on models up to 7B, the research question "Is explicit thinking always necessary and beneficial for RFT?" feels overly broad. The research question should be refined explicitly to address smaller-scale models (e.g., below 7B) or include additional experiments with larger models to strengthen the generality of the conclusion.
2. In some tasks, such as EuroSAT, the No-Thinking-RFT method underperforms compared to SFT. Do authors consider these cases as the outliers? Hope to see more insights and explanations regarding these discrepancies.
3. Do you think the Thinking-RFT has poor performance is because of the low-quality explicit thinking? Also, an open question is whether more fine-grained or nuanced reward designs for the thinking process could potentially improve Thinking-RFT’s performance. An ablation study or additional analysis would great to have.

Note: I'm happy to increase my score if the authors can address my concerns in weakness and questions sections.

**Ethical Concerns:**

["NO or VERY MINOR ethics concerns only"]

**Final Justification:**

I changed my score to 4 because the authors addressed most of my concerns and provided sufficient related work.

**Limitations:**

The experiments lack error bars, confidence intervals, or multiple runs with different random seeds to ensure robustness and statistical significance of the results.

**Paper Formatting Concerns:**

No formatting concerns.

**Quality:**

3

**Strengths And Weaknesses:**

Strengths:
- The paper extends the question of explicit thinking to multimodal LMs, providing valuable insight into scenarios where explicit thinking during RFT may not always be beneficial, particularly highlighting that for small models explicit thinking can sometimes negatively impact performance.
- The authors also introduce an Adaptive-Thinking approach, enabling the model to learn autonomously when explicit thinking should be applied, which is an interesting contribution.

Weaknesses:
- The authors should include a section in related work about the literature on the negative impacts of explicit thinking (e.g. cot & overthinking) This would provide necessary context and better frame the motivation for their proposed No-Thinking-RFT method. (e.g. https://arxiv.org/abs/2503.16419 (survey paper as a good reference), https://arxiv.org/abs/2410.21333 (task2)).
- The summary of contributions in the introduction is lengthy. It would be more effective and reader-friendly if presented in a concise and clear short sentences, highlighting the key contributions explicitly.
- The implementation details section needs more clarity and completeness. Important hyperparameters such as temperature, random seeds, and other settings are important for reproducing the results should be explicitly mentioned.
- Figures 2 and 3 have readability issues—the font size is too small, and the colors (particularly blue, representing decreases in Figure 2) are not clearly distinguishable. Authors should improve these visualizations to better show their results and enhance readability.
- Section 4 highlights several critical findings and contains substantial insights. This section might be better placed earlier in the paper (e.g., before the case study on classification tasks). Reorganizing the structure to prioritize main results and findings would enhance readability and clarity in presenting the core contributions.

---

> ### Author Rebuttal · Authors · 2025-07-31
>
> We sincerely appreciate your time and effort in reviewing our paper. We are encouraged that you found our insight into the explicit thinking process of RFT valuable. Below are our responses (R) to the weaknesses (W) and questions (Q) mentioned in your review.
>
> ------
>
> **W1: Related–work omits negative impacts of explicit thinking.**
>
> **R:** Thank you for the suggestion. We will add a subsection in *Related Work* that surveys the emerging literature on the drawbacks of chain-of-thought and thinking. We will be sure to cite and discuss *Stop Overthinking* [1] and *Mind Your Step* [2] in the related work of the next paper version. This will better frame the motivation for No-Thinking-RFT.
>
> ------
>
> **W2: Contribution summary is too long.**
>
> **R:** Thank you for the suggestion. We have written three more concise sentences to summarize our contributions and will include them in the next version of the paper.
>
> 1. We extend *Thinking-RFT* to few-shot MLLM classification and reveal a cross-dataset transfer across datasets.
> 2. We show several important findings about *No-Thinking-RFT* and Thinking-RFT via experiments on six tasks with 2B–7B models.
> 3. Through *Think-After-Answer* and *Adaptive-Thinking* we confirm that deferring or omitting CoT speeds convergence without harming accuracy and explore adaptive thinking strategy.
>
> ------
>
> **W3: Implementation details incomplete.**
>
> **R:** Due to space limitations, we briefly outline the implementation details in the main text and provide the full version in Appendix E. In the final version, we will move these details into the main text, including all parameter settings such as temperature (1.0 for all experiments) and random seed (100).
>
> ------
>
> **W4: Figures 2 and 3 are hard to read.**
>
> **R:** We apologize for the difficulty in readability. To improve clarity, we will enlarge the fonts, switch to a color-friendly “RdBu” palette, and use dotted outlines to highlight negative deltas for better contrast in the next version.
>
> ------
>
> **W5: Section 4 (findings) should appear earlier.**
>
> **R:** Thank you for the insightful suggestion. We will reorganize the paper by moving Section 4 earlier, beginning with the research questions, then presenting the main findings of Section 4, and finally proceeding to the task-specific case studies. This restructuring will allow readers to grasp the key results before delving into the detailed analyses.
>
> ------
>
> **Q1: Deepseek-R1 doesn’t prove thinking helps small models; yet your study stops at ≤ 7B, so “Is explicit thinking always necessary?” seems too broad.**
>
> **R:** **(i)** We would like to respectfully clarify that our motivation is *not* that Deepseek-R1 demonstrates benefits for small models, but that many recent multi-modal RFT works with 2B–7B models implicitly treat lengthy chain-of-thought as essential and try to reproduce the “length-increasing” effect (l. 34–36). Given the considerable time and cost of Thinking-RFT, we evaluate its necessity by exploring and comparing it to No-Thinking-RFT.
>
> **(ii)** Due to resource constraints, we cannot conduct experiments with larger model sizes such as 32B. We agree the original wording was overly broad and will revise the research question to sentence below in the next paper version.
>
> > *“Is explicit thinking always necessary and beneficial for **small-sized-model** RFT?”*
>
> ------
>
> **Q2: Explanation of why No-Thinking-RFT underperforms SFT on EuroSAT.**
>
> **R:** We view EuroSAT as an outlier since No-Thinking-RFT exceeds SFT on most datasets and by a large margin on average. The likely reason is the pre-training distribution of Qwen2-VL. EuroSAT consists of remote-sensing images, a domain we guess may be under-represented in the model’s pre-training corpus. Consequently, the model struggles to obtain positive rewards during RFT and the reward signals become extremely sparse, which impedes both Thinking- and No-Thinking-RFT learning processes. In contrast, SFT optimizes directly against ground-truth labels, avoiding the sparse-reward issue.
>
> ------
>
> **Q3: Thinking-RFT has poor performance because of the low-quality explicit thinking? More fine-grained reward designs for the thinking process could potentially improve Thinking-RFT’s performance.**
>
> **R: (i)** We attribute the weaker performance to the low quality of the explicit thinking. Evidence includes the much slower convergence of Thinking-RFT relative to No-Thinking-RFT (Figures 5 and 8), indicating that the reasoning steps impede learning, and the frequent inconsistencies between thoughts and final answers (Finding 3 in Sec. 4.3), which suggest confused or ill-formed reasoning.
>
> **(ii)** Finer-grained rewards could indeed improve performance. In Finding 3 we show that inconsistently tagged answers have lower accuracy than the overall average, implying that enforcing consistency may boost results. However, designing such rewards is non-trivial and challenging. A simple strategy is to use an auxiliary LLM to evaluate reasoning quality, but this approach is both time-consuming and susceptible to reward hacking, making it impractical.
>
> As an initial experiment, we designed a reward for the StanfordCars dataset that requires answers in the \<answer\> tag to also appear in the \<thinking\> tag; this preliminary approach yielded an almost 3-percentage-point boost in classification accuracy. Nonetheless, creating general, robust, and fine-grained reward functions and evaluating effectiveness remains challenging, so we leave that for future work.
>
> ------
>
> **Limitation about no error bars or multiple seeds.**
> **R:** We would like to clarify that, given the high cost of fine-tuning, we did not report error bars, which is also adopted in most previous MLLM and LLM training papers. We report the mean and standard deviation over three runs for both Thinking-RFT and No-Thinking-RFT using Qwen2VL-2B on CVBench. The results are as follows:
>
> - **Thinking-RFT:** 70.66 ± 0.30
> - **No-Thinking-RFT:** 76.93 ± 0.25
>
> These findings are largely consistent with those reported in the original paper.
>
> ------
>
> **References**
>
> [1] Stop Overthinking: A Survey on Efficient Reasoning for Large Language Models.
>
> [2] Mind Your Step (by Step): Chain-of-Thought can Reduce

---

> > ### Author Response · Authors · 2025-08-07
> >
> > Dear Reviewer wtZh,
> >
> > We would like to sincerely thank you for the thoughtful and constructive feedback. As we approach the end of the discussion period, we would like to ask you to kindly confirm that you have reviewed the rebuttal and let us know if there are any remaining concerns regarding our work.
> >
> > Thank you again for your help and time in reviewing our submission!
> >
> > Best Regards,
> >
> > All Anonymous Authors

---

> > ### Comment · Reviewer_wtZh · 2025-08-07
> >
> > Thank you for your response. The authors address most of my concerns, so I will consider to raise my score. However, I would expect author to do more literature review instead of just adding the two example papers that I provided.

---

> ### Author Response · Authors · 2025-08-07
> **More literature review about the negative impacts of overthinking**
>
> Thank you for your response. We apologize for not including sufficient literature in our initial rebuttal. Below, we provide a subsection discussing the overthinking problem, and we will include it in the next revision of the paper.
>
>
>
> **Overthinking in LLMs and MLLMs**
>
> Recent advancements in sophisticated reasoning abilities enabled by techniques such as Chain-of-Thought (CoT) prompting [1] have marked a significant milestone in the development of large language models (LLMs) and multimodal LLMs (MLLMs). CoT allows models to generate intermediate reasoning steps when solving complex problems, thereby improving both transparency and performance. However, this capability has also introduced a notable challenge referred to as the **"overthinking phenomenon"** [2].
>
> Overthinking describes the tendency of LLMs and MLLMs to produce unnecessarily verbose, redundant, and computationally expensive reasoning chains, even for simple queries. This behavior can hinder practical deployment and, in some cases, degrade performance [2–12]. For example, [3] shows that CoT can harm accuracy on tasks where extra deliberation impairs human performance. [4] quantifies overthinking in powerful LLMs and proposes pruning strategies, [5] shows excessive internal reasoning degrades success of LLM-based agents and [10] demonstrates that reasoning offers limited benefits on commonsense tasks. Additionally, [9] shows that overthinking can negatively impact MLLM inference accuracy and [12] enhances CoT reasoning by representing its steps in a continuous space which makes CoT more efficient and improve performance.
>
> While these studies provide valuable insights into model reasoning behavior, they focus exclusively on inference. The effect of explicit reasoning during reinforcement fine-tuning (RFT), however, remains largely unexplored.
>
>
> **References**
>
> [1] Chain-of-Thought Prompting Elicits Reasoning in Large Language Models, NeurIPS 2022.
>
> [2] Stop Overthinking: A Survey on Efficient Reasoning for Large Language Models, arXiv 2025.
>
> [3] Mind Your Step (by Step): Chain-of-Thought can Reduce Performance on Tasks where Thinking Makes Humans Worse, arXiv 2024.
>
> [4] Do NOT Think That Much for 2+3 = ? On the Overthinking of GPT‑4‑Style Models, ICML 2025.
>
> [5] The Danger of Overthinking: Examining the Reasoning‑Action Dilemma in Agentic Tasks, arXiv 2024.
>
> [6] When More is Less: Understanding Chain-of-Thought Length in LLMs, Reasoning and Planning for LLMs @ ICLR2025.
>
> [7] Unveiling Confirmation Bias in Chain-of-Thought Reasoning, ACL 2025 Findings.
>
> [8] Are Machine Rationales (Not) Useful to Humans? Measuring and Improving Human Utility of Free-Text Rationales, ACL 2023.
>
> [9] Mme-cot: Benchmarking chain-of-thought in large multimodal models for reasoning quality, robustness, and efficiency, ICML 2025.
>
> [10] To cot or not to cot? chain-of-thought helps mainly on math and symbolic reasoning, ICLR 2025.
>
> [11] C3oT: Generating Shorter Chain-of-Thought without Compromising Effectiveness, AAAI 2025.
>
> [12] SoftCoT: Soft Chain-of-Thought for Efficient Reasoning with LLMs, ACL 2025.

---

### Official Review · Reviewer_jHwD · 2025-07-02

**Clarity:** 3
**Significance:** 4
**Originality:** 3
**Rating:** 4
**Confidence:** 4

**Summary:**

The paper proposes No-Thinking-RFT, replacing the reward of RFT with an equality accuracy award. The No-Thinking-RFT is compared against RFT first in an image classification task, followed by a suite of visual reasoning tasks. Further analyses, including Think-After-Answer and Adaptive-Thinking are included to strengthen the paper's comprehensive analysis on thinking in RFT. Overall findings suggest that some tasks benefit from RFT, but often times No-Thinking-RFT is better --- creating a new paradigm to consider in training large multimodal models.

**Questions:**

In addition to weaknesses, I have a few questions:
- Were the datasets reported in the paper the only datasets tested? If so, is there a pre-registration (i.e., on OSF) to indicate this? This would greatly increase the findings.
- It would be good to discuss your results a bit more via a discussion section. Currently, it is a bit hard to contextualize the results of the work into the existing research landscape. It is also necessary to include a discussion of limitations and future directions to help support future work.

**Ethical Concerns:**

["NO or VERY MINOR ethics concerns only"]

**Final Justification:**

During the rebuttal period, the authors have improved the paper writing significantly, contextualizing their findings to a broader audience. While there are still some limitations to the breadth of tasks evaluated, limiting the score, these cannot reasonably be completed within the limited time frame of the rebuttal. I consider all my points addressed and support acceptance.

**Limitations:**

No - based on my reading, I did not find any limitations discussed in the paper. While there is limited negative societal impact for such technical work, one idea could be that if we remove thinking, the model traces could be less interpretable, or model behavior could potentially be more brittle given the more fixed equality training metric.

**Quality:**

3

**Strengths And Weaknesses:**

Strengths:
- Timely paper on up to date methodologies.
- Analyses seem thorough. I especially appreciated the think-after-answer and adaptive-thinking experiments, which add validity to the authors' claims.
- Explanation for why the convergence is faster and why performance increases both make a lot of sense; creating a cohesive and believable set of takeaways.

Weaknesses:
- Framing could be stronger in the introduction and (especially) the abstract. I think the writing lacks motivation for why the problem is important.
- Some missing citations: On overthinking harming performance [1], on adaptive selection of thinking/no thinking [2].

[1] Liu, R., Geng, J., Wu, A. J., Sucholutsky, I., Lombrozo, T., & Griffiths, T. L. (2024). Mind your step (by step): Chain-of-thought can reduce performance on tasks where thinking makes humans worse. ICML 2025.
[2] De Sabbata, C. N., Sumers, T. R., & Griffiths, T. L. (2024). Rational metareasoning for large language models. arXiv preprint arXiv:2410.05563.

---

> ### Author Rebuttal · Authors · 2025-07-31
>
> We sincerely appreciate your time and effort in reviewing our paper. We are encouraged that you found our analysis both thorough and cohesive. Below are our responses (R) to the weaknesses (W) and questions (Q) mentioned in your review.
>
> ------
>
> **W1: Framing could be stronger in the introduction and (especially) the abstract.**
>
> **R:** We appreciate this observation and will strengthen both the abstract and Introduction section by foregrounding the practical gap that motivates our study, drawing on claims below:
>
> - **Why the question matters.** Rule-based reinforcement fine-tuning (RFT) has recently surpassed supervised fine-tuning (SFT) on a range of tasks (l. 32–34), yet it is *computationally expensive* because it generates multiple long responses (l. 41–43). If explicit thinking is *not* always necessary (l. 36–40), practitioners could reduce latency, GPU hours, and energy costs without sacrificing accuracy.
> - **Untested assumption.** Explicit chain-of-thought is widely assumed to be the key to RFT’s success (l. 34–36), but prior work showing “overthinking” harms performance focuses only on inference-time reasoning, not on the **training** process (l. 38–40). Whether thinking is beneficial *during RFT* therefore remains unclear.
> - **Our contribution.** We introduce *No-Thinking-RFT*, a version of RFT that removes the thinking step by using a strict equality reward and direct-answer prompts (l. 60–64). Across tasks and model sizes, this variant outperforms Thinking-RFT while converging faster and requiring shorter fine-tuning and inference (l. 64–65), thus addressing both accuracy and efficiency.
>
> ------
>
> **W2: Some missing citations: On overthinking harming performance [1], on adaptive selection of thinking/no thinking [2].**
>
>  **R:** Thank you for these valuable references. We will add a subsection in the related work to discuss prior studies on overthinking, and we will be sure to cite these two papers in the next version.
>
> ------
>
> **Q1: Were the datasets reported in the paper the only datasets tested? If so, is there a pre-registration (i.e., on OSF) to indicate this? This would greatly increase the findings.**
>
>  **R:** Yes, the datasets reported in the paper are the only ones we tested. Although we did not carry out a formal *pre-registration*, every dataset choice strictly follows established practice in prior works rather than post-hoc selection. For example:
>
> - **Few-shot classification:** we follow *CoOp* [1] and use the standard suite of eleven datasets for training and evaluation; this protocol is adopted by nearly all recent papers in few-shot image classification.
> - **Spatial understanding:** we follow *VisualThinker-R1-Zero* [2], training on the SAT dataset and testing on CVBench.
> - **Mathematical VQA:** we follow *Math-LLaVA* [3], training on Math-40K and evaluating on the MathVista and MathVision datasets.
>
> No additional datasets were tried or discarded and we purely followed the experimental setting in previous works to conduct experiments for our study. To increase transparency, we will create an OSF record that time-stamps the complete experimental protocol.
>
> ------
>
> **Q2: It would be good to discuss your results a bit more via a discussion section. Currently, it is a bit hard to contextualize the results of the work into the existing research landscape.**
>
>  **R:**  Thank you for the suggestion. We will reorganize the manuscript so that, after introducing the research questions that motivate the study, the experimental sections will present the results, followed by a dedicated **Discussion** section that interprets how the findings answer each question. In particular, we plan to discuss the following topics/questions (here we provide a short discussion section; we will provide a more lengthy discussion in the next version):
>
> 1. **Are longer chain-of-thought reasoning processes always beneficial?**
>    Quite a few recent works on multimodal RFT have tried to reproduce the “length-increasing” effect (l. 34-36) originally reported in Deepseek-R1, treating lengthy chain-of-thought traces as essential for multimodal problem solving. However, our results show this is not always true. As *finding 1* indicates (l. 258), the model’s reasoning is often trivial—adding little to the final answer—especially on complex tasks when capacity is limited (2 B). Thus, the token budget devoted to CoT may be better spent on answer tokens or larger batch sizes when model capacity is tight.
>
> 2. **Does model size mediate the gains of thinking- vs. no-thinking-RFT?**
>    Larger models are less prone to incoherent reasoning, improving thinking-RFT results. Yet, as Tables 3 & 4 and *finding 2* show, no-thinking-RFT still outperforms thinking-RFT on vision-centric and puzzle-related tasks, while generally lagging on math tasks—regardless of model size. Hence, although capacity influences each method’s absolute performance, the overall trend remains: task type largely determines which variant excels.
>
> 3. **When should we choose thinking-RFT over no-thinking-RFT (and vice-versa)?**
>    Choice depends on task, budget, and requirements. Vision-centric and puzzle tasks benefit more from no-thinking-RFT, whereas reasoning-heavy tasks like math gain from thinking-RFT. Because it skips explicit reasoning, no-thinking-RFT runs faster and is more compute-friendly. Conversely, when explicit reasoning aids performance—e.g., in medical reasoning—thinking-RFT is preferable.
> ------
>
> **Q2 & Limitation: It is also necessary to include a discussion of limitations and future directions to help support future work.**
>
>  **R:** Thank you for the suggestion. Due to page limit, we placed the limitation section in Appendix A.1. We will include the limitation section in the main body of the paper and provide a more thorough discussion by incorporating the reviewers’ comments in the next version.
>
> ------
>
> **References**
>
> [1] Learning to Prompt for Vision-Language Models, IJCV 2022
>
> [2] Math-LLaVA: Bootstrapping mathematical reasoning for multimodal large language models, EMNLP 2024
>
> [3] R1-Zero’s “Aha Moment” in Visual Reasoning on a 2B Non-SFT Model, arXiv 2025

---

> > ### Author Response · Authors · 2025-08-07
> >
> > Dear Reviewer jHwD
> >
> > We would like to sincerely thank you for the thoughtful and constructive feedback. As we approach the end of the discussion period, we would like to ask you to kindly confirm that you have reviewed the rebuttal and let us know if there are any remaining concerns regarding our work.
> >
> > Thank you again for your help and time in reviewing our submission!
> >
> > Best Regards,
> >
> > All Anonymous Authors

---

> ### Comment · Reviewer_jHwD · 2025-08-07
>
> Thank you for the rebuttal. I consider all my points addressed. I especially appreciate the improved introduction and discussion points, as well as the OSF record. While my score is already positive, I am happy to increase my confidence that the paper should be accepted. I have also increased my clarity score.

---

> > ### Author Response · Authors · 2025-08-08
> >
> > We sincerely thank you for your recognition of our work and your constructive comments.

---

### Official Review · Reviewer_2mLo · 2025-07-02

**Clarity:** 3
**Significance:** 2
**Originality:** 2
**Rating:** 5
**Confidence:** 4

**Summary:**

The paper investigates the usefulness of ‘thinking’ during RL training for small MLLMs (sizes 2B, 7B). Their main findings are (a) thinking during RL finetraining outperforms SFT (b) however, thinking actually seems to hinder training. Instead, doing direct RL on actions (without thinking) is more sample efficient and requires considerably less wall-time  (c)  They find that adaptive thinking RL can perform marginally better (compared to No-Thinking RL). On some reasoning tasks, like mathematical problems, they do observe a gain for Thinking RL.

**Questions:**

1. How likely is that the model is only learning to ‘format’ the answer correctly to be parsable by the chosen parser than learning new capabilities?
2. Reasoning training on small LLMs often requires some sort of warmup; did the authors try any sort of SFT warmup; or generally provided any curriculum for thinking?

**Ethical Concerns:**

["NO or VERY MINOR ethics concerns only"]

**Final Justification:**

Authors provided a comprehensive rebuttal with some results investigating why no-thinking RFT outperforms thinking-RFT. Specifically, 'Analyzing which model parts change during RL.' part of rebuttal I believe presents interesting findings.

**Limitations:**

Limitations should be in the main paper, and more thorough.

**Quality:**

3

**Strengths And Weaknesses:**

### Strengths

1. Paper is well-written.
2. Experiments are conducted across range of datasets.
3. Paper has done a relatively ok job of disentangling some key hypotheses through creative experimentation.

### Weaknesses

Note to authors: Weaknesses are important to address. Either through including novel results, and/or including relevant discussion (e.g., in limitations), and/or convincing the reviewer that their critique is faulty. Improvements are things that I believe would be helpful to improve the paper but are not necessary to address (though addressing enough of them could mean I improve my score).

1. The paper’s limitations section does not acknowledge its limitations in satisfactory way, and ought to be included in the main body.
2. The image classification datasets used are all highly popular ones. The paper does not discuss the ramifications of this. The dataset memorization could explain why no-thinking does better. I would ideally like the paper to test on novel synthetic image datasets (with potentially novel classes / objects) to mitigate this risk. Or if the authors don't think this is a risk, I would like to hear their explanation.
3. Image classification experiments results in the main paper are only presented for 2B model. 7B model results should also be in the main paper (currently table 10 in appendix) and done for all models. It seems gap shrinks significantly for 7B models between thinking / no-thinking RL (on OxfordFlowers it goes from 5 points to \<.5 points).
4. The image classification setup reduces the number of classes. I imagine the number of classes as potentially a factor in model performance and would be interested in some analysis of this aspect. Would be interesting to understand how \# of classes impacts model performance. Closest experiment to this is open-set classification experiments with results in Table 9 in which the gap shrinks significantly.
5. I am somewhat concerned that the RL training setup used in the paper is not optimal; no SFT warmup is used, no curriculum is used etc. So, another plausible explanation of the paper’s results could be that RL training was not done well. It is somewhat surprising to me that on reasoning tasks, thinking does not help (on classification tasks, it sort of makes sense that reasoning would be a distraction considering it is a recall based task).
6. Related to 5: 	Line 274-276 present two interesting hypothesis to explain why 7B model does not benefit from thinking on puzzle tasks. It would be interesting to try and disentangle which hypothesis is true. One way could be to give the LLM tools to perform visual reasoning in-context (e.g., zoom / crop etc.) and see if that helps model’s performance improve. “This may be because these tasks rely on visual rather than linguistic reasoning, and language-based reasoning might cause hallucinations, or the 7B model’s puzzle-solving capability may still be limited.”

### Improvements

1. The work provides prompts for all the settings which is good. It would be good to collect them within a single appendix section (instead of being spread throughout the appendix).
2. Create a distinct subsection in appendix where details of all datasets (and ideally one/two samples from the dataset) are provided. Some of the datasets were novel to me.
3. The appendix has many interesting results; would be helpful if authors could include some sort of table of content and preamble to the appendix to help readers navigate it better.
4. Doing some sort of analysis of what model parts change through the RL training would be insightful; e.g., are the changes concentrated in the vision encoder part or the LLM part?
5. Most of the results in the paper are presented in the form of table; it might be better to present them in terms of spider chart / radar plot where feasible. This presentation would be more amenable to a quick skim.

---

> ### Author Rebuttal · Authors · 2025-07-31
>
> Thank you for your  thoughtful feedback. We truly appreciate your suggestions for improving paper. Since we cannot revise paper during rebuttal, we will incorporate your comments in the next version. Below are our responses (R) to the weaknesses (W), improvement (I) , and questions (Q).
>
> **W1: The limitation section should be better addressed in the main text.**
>
> **R:** Due to page limitations, we placed it in the appendix. We will include this in the main text and provide a more thorough limitation discussion by incorporating the reviewers' comments in the next version.
>
> ---
>
> **W2: Dataset memorization risks; test on novel synthetic data.**
>
> **R:** We agree that contamination is a legitimate concern. Following your suggestion, we used GPT-4 to create two *synthetic class groups* with novel class names absent from existing datasets, each accompanied by a brief GPT-4-generated description:
>
> - **Fantasy objects** e.g. *Frostleaf*:  Serrated emerald leaf edges coated in prismatic rime-ice crystals
> - **Mathematical shapes** e.g. *Ellipse*: horizontal ellipse outline centered on a plain background
>
> Then using FLUX‑1‑dev we synthesized 4 training and 20 test images per class (30 classes total) and fine‑tuned Qwen2VL‑2B. The results are shown below. No‑Thinking‑RFT remains decisively ahead on both synthetic datasets, indicating that its advantage is **not** due to memorization.
>
> | Model           | Fantasy   | Math      |
> | --------------- | --------- | --------- |
> | Qwen2VL-2B      | 15.5      | 30.99     |
> | Thinking-RFT    | 30.99     | 43.33     |
> | No-Thinking-RFT | **55.33** | **49.33** |
>
> ---
>
> **W3: Only 2B results are shown in main text; 7B results in appendix should be included.**
>
> **R:** Due to page limit, we included 7B model classification results in the appendix. We provide a more detailed comparison of 7B model below and will include this table into the main text.
>
> | Method             | DTD       | EuroSAT   | Flowers   | S.C.      | Average   |
> | ------------------ | --------- | --------- | --------- | --------- | --------- |
> | Thinking-RFT       | 77.90     | 53.17     | 93.91     | 84.19     | 77.29     |
> | Think-After-Answer | 76.29     | **62.95** | 94.84     | 89.32     | 80.85     |
> | Adaptive-Thinking  | 79.60     | 56.62     | **96.86** | 89.92     | 80.75     |
> | No-Thinking-RFT    | **80.56** | 58.91     | 94.24     | **94.02** | **81.93** |
>
> ---
>
> **W4: Study how the number of classes affects performance**
>
> **R:** The number of class names in the choice list reflects task difficulty, as models perform better with fewer options. We tested varying sizes on StanfordCars (20, 40, 60, 78 [# in paper], 100) and UCF-101 (20, 40 [# in paper], 60, 80, 100). The results are shown below. Fewer class choices boost accuracy for both models and shrink the Thinking- vs. No-Thinking gap.
>
> | Dataset | Model           | 20    | 40    | 60    | 78 / 80 | 100   |
> | ------- | --------------- | ----- | ----- | ----- | ------- | ----- |
> | DTD     | Thinking-RFT    | 85.60 | 69.92 | 65.23 | 63.42   | 57.92 |
> |         | No-Thinking-RFT | 86.53 | 73.52 | 70.79 | 70.15   | 65.33 |
> | S.C.    | Thinking-RFT    | 96.52 | 93.78 | 86.24 | 80.24   | 80.59 |
> |         | No-Thinking-RFT | 97.16 | 94.70 | 93.91 | 92.50   | 90.88 |
>
> ---
>
> **W5: concerns about RL setup, without SFT warmup or curriculum.**
>
> **R:** We agree that the training setting (Zero-RL) might not be optimal. However, our aim is to study thinking in RFT, not to introduce a new SOTA algorithm. For this purpose, we believe Zero-RL is the most appropriate choice, for the following reasons:
>
> - **Focus on the fundamental contrast.** Zero-RL provides the cleanest environment to study the core question—*The role of thinking in RFT across different model sizes (2/7 B)*—without additional factors that could mask or amplify either side.
> - **Precedent and accessibility.** Many prior works adopt zero-RL (e.g., MMEureka, VisualThinker-R1-Zero, and VLM-R1); repeating that setup ensures our findings are directly comparable and reproducible across domains.
> - **Variability of CoT-SFT data.** CoT SFT hinges on data quality; uneven or low-quality corpora can even decrease results, while high-quality data are not universally obtainable. Relying on them would limit the scope of the study.
> - **SFT is a capability booster.** High-quality SFT boosts capability much like scaling model. To study thinking across different ability levels, it is sufficient to vary model size.
> - **Curriculum learning (CL) reduces generality.** Findings tied to a specific CL may not generalize, and different curricula can produce diverging results. Zero-RL removes this variance, making our conclusions broadly applicable.
>
> Accordingly, we adopt Zero-RL as our training paradigm, consistent with other RFT studies [1].
>
> For completeness, we ran two small ablations on MathVista: a SFT warm-up and a CL schedule.
>
> **1. SFT warm-up**
>  We generated 20k CoT data with GPT-4o to fine-tune 2B model, then trained both Thinking and No-Thinking-RFT from this SFT checkpoint. Results are below. With SFT warm-up, both methods improve, but No-Thinking-RFT still leads.
>
> | Model           | w/ SFT | w/o SFT |
> | --------------- | ------ | ------- |
> | Thinking-RFT    | 45.52  | 44.90   |
> | No-Thinking-RFT | 49.21  | 48.80   |
>
> **2. Curriculum learning (CL)**
> We generated 10 outputs per question using Qwen2VL-2B and assigned difficulty based on average accuracy: easy (0–0.33), medium (0.33–0.66), hard (0.66–1.0). CL stages: Stage 1 = easy, Stage 2 = medium, Stage 3 = hard
>
> This CL strategy even yields a slightly lower result for Thinking-RFT: 44.46 (w/ CL) vs. 44.90 (w/o CL).
>
> ---
>
> **W6: Use visual-context fine-tuning (FT) to determine which explanation accounts for limited puzzle gains.**
>
> **R:** We appreciate this suggestion. However, visual-context FT experiments are outside the scope of our paper on RFT thinking. Given limited rebuttal time, we cannot implement it effectively.
>
> As a pragmatic proxy, we evaluated the o4-mini model, whose API supports optional in-context visual reasoning (e.g., cropping). On PuzzleVQA, performance rises from 84.30% (no visual reasoning) to 86.48% (with visual reasoning), a gain of +2.18% accuracy. This suggests that visual-context learning can improve puzzle performance.
>
> We will add this result to the paper, acknowledge that tool-using FT for puzzleVQA as an limitation and leave it to future work.
>
> ---
>
> **I1: Group all prompts in a single appendix section.**
>
>  **R:** We will collect the prompts in Appendix E.4 and G into a single section in the next version.
>
> ---
>
> **I2: Add a appendix subsection with dataset details**
>
> **R:** We will consolidate the dataset introduction sections in the appendix into a single unified section, and will include several representative samples from each dataset.
>
> ---
>
> **I3: A table of contents and preamble in the appendix.**
>
>  **R:** We will add an organisation guide section and a table of content and preamble to the appendix in the next version.
>
> ---
>
> **I4: Analyzing which model parts change during RL.**
>
> **R:** This is an insightful point. We compare the changes by computing the L₂ norm of parameter difference. Our analysis focuses on three aspects:
>
> 1. **Modality**: changes between visual and language components
> 2. **Module**: changes in different modules
> 3. **Layer**: changes across different layers
>
> We analyzeQwen2VL-2B on SAT, DTD, and Math datasets using Thinking and No-Thinking-RFT. Since we cannot upload images, Modality- and module-level results are given in the table below. We directly discuss layer-level findings as the table is too large.
>
> (i) **Modality-Level:** Language weights drift more than visual weights for all dataset—and the No-Thinking strategy raises visual drift slightly. This implies that the reward gradient mainly applies to the language modality.
>
> (ii) **Module-Level:** MLP is the dominant changed component for both visual and language parts, and attention blocks change less but still contribute near 15–20%.
>
> (iii) **Layer-Level:** With Thinking, weight drift grows toward deeper layers as reward back-propagates through the whole reasoning chain to high-level semantics. With No-Thinking, drift peaks in early-mid layers and then declines, indicating that low-level features are reshaped so a shallow forward path already produces the reward token.
>
> | Dataset | Method      | Visual | Language | MLP_V | norm_V | attn_V | MLP_L | norm_L | attn_L |
> | ------- | ----------- | ------ | -------- | ----- | ------ | ------ | ----- | ------ | ------ |
> | DTD     | Thinking    | 0.267  | 0.434    | 0.206 | 0.001  | 0.158  | 0.372 | 0.000  | 0.153  |
> |         | No-Thinking | 0.237  | 0.402    | 0.184 | 0.001  | 0.143  | 0.379 | 0.000  | 0.154  |
> | SAT     | Thinking    | 0.503  | 0.712    | 0.398 | 0.003  | 0.311  | 0.614 | 0.000  | 0.256  |
> |         | No-Thinking | 0.518  | 0.718    | 0.398 | 0.003  | 0.327  | 0.704 | 0.000  | 0.281  |
> | Math    | Thinking    | 0.415  | 0.735    | 0.364 | 0.002  | 0.210  | 0.691 | 0.000  | 0.270  |
> |         | No-Thinking | 0.731  | 0.931    | 0.581 | 0.004  | 0.455  | 0.880 | 0.000  | 0.388  |
>
> ---
>
> **I5: Consider using radar plots instead of tables where feasible.**
>
> **R:** We will use spider chart and radar plot to present results such as Table 3, 4, and 5 in the next version to make the paper more amenable.
>
> ---
>
> **Q1: Is the model just learning to format rather than gaining new capabilities?**
>
> **R:** Whether RFT teaches new skills is an open problem. Our models outperform zero-shot baselines, suggesting real learning. [1] finds RFT boosts pass@1 while leaving high-k pass@k unchanged, hinting it may just rank the correct answer first.
>
> ---
>
> **Q2: SFT warmup or any curriculum for small-model training?**
>
> **R:** We ran a brief SFT warm-up and a CL schedule ablation; see W5 response.
>
> ---
> **References**
>
>  [1] Does reinforcement learning really incentivize reasoning capacity in llms beyond the base model? arXiv 2025.

---

> > ### Comment · Reviewer_2mLo · 2025-08-03
> >
> > Thanks for your response. I have increased my rating from 4 to 5. This was primarily triggered by the rebuttal part on 'Analyzing which model parts change during RL.' I think this is interesting and perhaps should be more salient in final version.

---

> > > ### Author Response · Authors · 2025-08-04
> > > **Thank you for increasing rating**
> > >
> > > Thank you for kindly increasing your rating. We sincerely thank you again for your valuable suggestions to improve our paper. We will make sure to include additional results on parameter changes in the final version, visualize the changes in figure form for better understanding, and discuss them more thoroughly in final version.

---

### Decision · Program_Chairs · 2025-09-17

**Decision:**

Accept (spotlight)

**Comment:**

This paper analyzes the effects of thinking in RL fine-tuning of MLLMs. The paper shows that RL outperforms SFT, yet thinking can often hinder final performance. The paper is a timely close look at the effects of thinking in MLLMs, and for practitioners has high utility. There were some weaknesses wrt ablations and datasets as well as some writing concerns, but a vast majority of these appear to have been addressed very well.

I therefore recommend the paper is accepted for spotlight.